# Recreating the synthesis of starch granules in yeast

Barbara Pfister[1], Antoni Sánchez-Ferrer[2], Ana Diaz[3], Kuanjen Lu[1], Caroline Otto[1], Mirko Holler[3], Farooque Razvi Shaik[3], Florence Meier[1], Raffaele Mezzenga[2], Samuel C Zeeman[1]*

[1]Department of Biology, ETH Zürich, Zürich, Switzerland; [2]Health Sciences and Technology, ETH Zürich, Zurich, Switzerland; [3]Paul Scherrer Institut, Villigen, Switzerland

**Abstract** Starch, as the major nutritional component of our staple crops and a feedstock for industry, is a vital plant product. It is composed of glucose polymers that form massive semi-crystalline granules. Its precise structure and composition determine its functionality and thus applications; however, there is no versatile model system allowing the relationships between the biosynthetic apparatus, glucan structure and properties to be explored. Here, we expressed the core Arabidopsis starch-biosynthesis pathway in *Saccharomyces cerevisiae* purged of its endogenous glycogen-metabolic enzymes. Systematic variation of the set of biosynthetic enzymes illustrated how each affects glucan structure and solubility. Expression of the complete set resulted in dense, insoluble granules with a starch-like semi-crystalline organization, demonstrating that this system indeed simulates starch biosynthesis. Thus, the yeast system has the potential to accelerate starch research and help create a holistic understanding of starch granule biosynthesis, providing a basis for the targeted biotechnological improvement of crops.

*For correspondence: szeeman@ethz.ch

**Competing interests:** The authors declare that no competing interests exist.

## Introduction

Starch is an agricultural raw material coveted for its nutritional value, but also for its functional properties, which have extensive applications in the food industry, in paper manufacturing, and in the production of biodegradable materials, amongst others (*Ellis et al., 1998*; *Blennow et al., 2003*; *Zhang et al., 2014*). Furthermore, special starches with reduced digestibility (resistant starch) are considered to be health-promoting, having a lower glycemic index and serving as a form of dietary fiber (*Raigond et al., 2014*).

Both industrial use and nutritional benefits depend on distinct biophysical characteristics and therefore on the underlying structure of starch. Starch is comprised of two α-polyglucan components (here named glucans): large moderately-branched amylopectin molecules and interspersing small, near-linear amylose molecules (*Manners, 1989*). Both are built from chains of α-1,4 linked glucose units that are connected via α-1,6-linkages (branch points). In amylopectin, the major and essential component, the branching pattern is non-random, giving rise to clusters of unbranched chain segments (*Pérez and Bertoft, 2010*). Neighboring chains within these clusters form double helices that pack with distinct inter-helical spacing. The resulting crystalline lamellae, alternating with amorphous lamellae containing the branch points, stack with a periodicity of ~9–10.5 nm (*Jenkins et al., 1993*). This unique semi-crystalline nature of amylopectin renders starch insoluble.

Amylopectin biosynthesis involves multiple enzyme activities, often composed of several isoforms with distinct specificities: four classes of starch synthases (SSs) elongate glucan chains using ADPglucose as glucosyl donor, creating α-1,4 glucose linkages; two classes of branching enzymes (BEs) introduce branches in the form of α-1,6 linked chains; and at least one isoamylase-type debranching

**eLife digest** Most plants and algae produce a carbohydrate called starch, which provides the plant with a dense store of energy. Starch is also the main carbohydrate in our diet and its unusual physical properties mean that it has many industrial uses. It is made of two different sugar-based molecules known as glucans and forms large, partially crystalline granules inside plant cells. Several enzymes are known to be involved in making starch, yet it is not clear exactly how the process works.

Animals and fungi cannot make starch but they do make another type of carbohydrate called glycogen, which is also a glucan. Yeast is a single-celled fungus that is often used in research because it is easy to genetically engineer and quick to grow. To study the plant enzymes that make starch in more detail, Pfister et al. aimed to genetically engineer yeast to make their own starch.

For the experiments, different combinations of enzymes involved in starch production in a plant called *Arabidopsis thaliana* were inserted into mutant yeast cells that were unable to make glycogen. The experiments show that all the plant enzymes are active in yeast and retain the roles that they perform in plants. Some of the enzyme combinations yielded glucan granules that occupied a large part of the yeast cell. These granules had many of the physical characteristics of plant starch, showing that yeast can be used as a system to better understand how starch is made.

Important next steps will be to insert more plant proteins into the yeast and to fine-tune the production of these proteins. This should help researchers to design starches with desired properties in yeast and ultimately engineer crop plants to produce them.

enzyme hydrolyzes some branches again, probably tailoring the glucan for crystallization (*Pfister and Zeeman, 2016*; *Tetlow and Emes, 2011*; *Zeeman et al., 2010*; *Jeon et al., 2010*). In contrast, amylose is synthesized by a single class of granule-bound starch synthases (*Denyer et al., 2001*). According to the classification of carbohydrate-active enzymes (CAZy; *Lombard et al., 2014*), all SSs belong to the glycosyltransferase family 5 (GT5), while BEs and isoamylases belong to distinct subfamilies of the glycoside hydrolase family 13 (GH13).

Starches from different plants vary in terms of amylose content, amylopectin structure and glucan modifications (e.g. level of phosphorylation) (*Santelia and Zeeman, 2011*). However, to date, this variation cannot fully meet industrial needs, necessitating costly physicochemical modifications post-extraction to yield the desired properties (*Tharanathan, 2005*). Although the starch-biosynthetic enzymes are highly conserved, we still do not have sufficient knowledge required for the rational modification of starch structure and properties in crops. This deficiency has several roots; first, the difficulty in generating plant mutants hinders a systematic analysis in most species. Second, the results obtained from comparable mutants in different plant systems are not always identical (e.g. due to variation in genetic, environmental and/or developmental backgrounds). Third, enzyme kinetics obtained from soluble substrates in vitro may not be representative of how an enzyme acts on a crystallizing surface in vivo (*O'Neill and Field, 2015*). These problems are further compounded by the fact that the glucans that make up starch are polydisperse, necessitating multiple biochemical and biophysical techniques for their structural characterization. Collectively, these limitations impair our ability to define enzyme functions unambiguously and to derive an overall model of starch biosynthesis. As a result, progress in engineering new starches with enhanced functionalities *in planta* has been empirical and slow, and the full potential of this important renewable resource has not been realized.

Here we created a model system in yeast that allows the expression of multiple starch-biosynthetic enzyme combinations in a targeted, controlled and fast manner. Using a suite of molecular and biochemical analyses, we show that the enzymes are functional and, in the right combinations, capable of producing semi-crystalline starch-like granules. This shows that our yeast system is a highly tractable experimental system in which starch biosynthesis can be simulated and the capacities of different interdependent enzymatic combinations can be evaluated, bringing us much closer to a holistic view of this critical process.

## Results

### Establishment of the yeast system

Our yeast system employs an expression platform in haploid *Saccharomyces cerevisiae* CEN.PK113-11C, which had been developed for stable heterologous gene expression (*Mikkelsen et al., 2012*). To eliminate interfering effects from the yeast's endogenous glycogen metabolic pathway, we purged it of the five genes involved in glycogen biosynthesis (two glycogenins, two glycogen synthases and glycogen branching enzyme) and two involved in glycogen degradation (glycogen debranching enzyme and α-glucan phosphorylase; *Figure 1*). We simultaneously introduced all known amylopectin-biosynthetic genes from *Arabidopsis thaliana* via stable integration of coding sequences driven by galactose-inducible promoters. These are the starch synthases *SS1* (AGI locus code At5g24300), *SS2* (At3g01180), *SS3* (At1g11720) and *SS4* (At4g18240), the branching enzymes *BE2* (At5g03650) and *BE3* (At2g36390) and the isoamylase encoded by *ISA1* (At2g39930) and *ISA2* (At1g03310) (see *Figure 1—figure supplement 1* for details on constructs). ISA1 and ISA2 together form the active heteromultimer that here we name ISA. Furthermore, we tested *BE1* (At3g20440), a third Arabidopsis gene annotated as branching enzyme (*Han et al., 2007*). This gene is distantly related to known branching enzymes, but studies to date have not revealed any starch-related func-

tion (*Dumez et al., 2006*; *Wang et al., 2010*). Arabidopsis was chosen as the donor for amylopectin-biosynthetic genes because it has been intensively used for starch research, resulting in well-characterized mutants that enable us to make direct comparisons between yeast- and plant-derived glucans. In addition, we introduced a non-regulated form of ADPglucose pyrophosphorylase from *Escherichia coli* (GlgC-TM; *Sakulsingharoj et al., 2004*), as yeast glycogen biosynthesis is UDPglucose dependent, while the plant starch biosynthesis is ADPglucose dependent.

We first established that the heterologously expressed enzymes are functional. Zymograms of soluble native protein extracts from yeast strains expressing ISA and single isoforms of SS and BE showed that each enzyme was expressed and active (SS1: lines 0 and 1; SS2: lines 2 and 3; SS3: line 5; SS4: line 7; BE2: line T; BE3: line S; ISA: all lines with uneven numbers; *Figure 2A*). The enzymes migrated similarly as in extracts of the wild-type (WT) Arabidopsis plant controls, even though we noted additional activity bands in some cases, possibly due to small alterations in post-translational modifications. Occasionally, we observed that protein activity varied in strength between the lines (e.g. SS2 activity between lines 2 and 3). We reasoned that the enzyme's binding to insoluble glucans potentially accumulating in these lines may affect its extractability as a soluble protein. Indeed, protein abundance detected in Western blots using total protein extracts – where proteins in the insoluble fractions had been solubilized by boiling in SDS-containing buffer – was more uniform (*Figure 2B*), suggesting that expression levels

Figure 1. Workflow of the *S.cerevisiae* system. The yeast's endogenous glycogen-metabolic pathway (grey box) was removed and varying components of the starch-biosynthetic pathway from Arabidopsis (green box) was introduced. The deregulated mutant of ADPglucose pyrophosphorylase (AGPase) from *E. coli* was used for the supply of ADPglucose. For details of the constructs see *Figure 1—figure supplement 1*.
The following figure supplement is available for figure 1:

**Figure supplement 1.** Constructs for heterologous gene expression and/or deletion of endogenous genes.

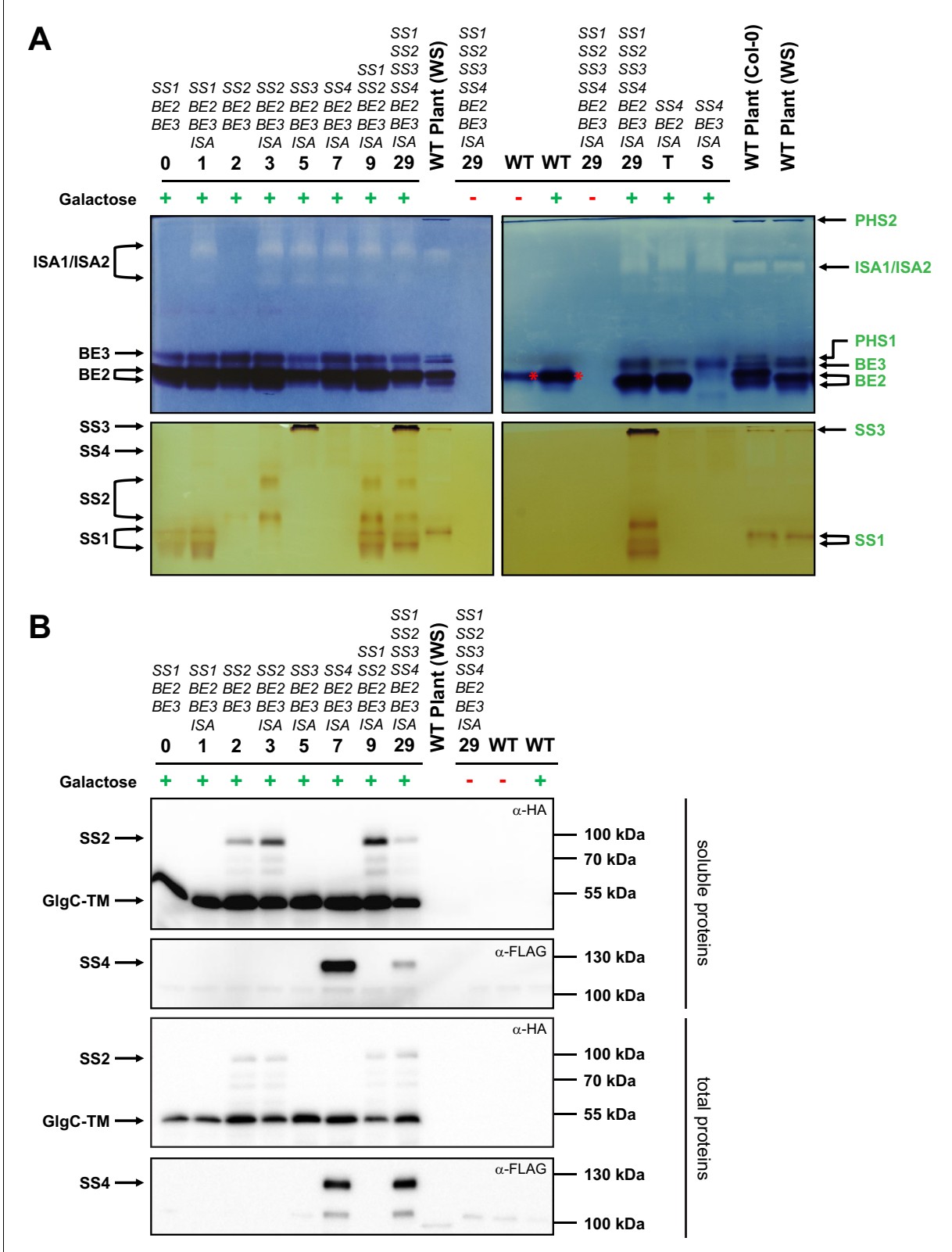

**Figure 2.** Native PAGE and immunoblots of heterologously expressed proteins. Total proteins or native soluble proteins were extracted from yeast lines grown in liquid cultures with complex medium. Native soluble proteins were subjected to native PAGE for detection of enzyme activity (**A**) and SDS-PAGE followed by Western blotting (**B**, upper panels). Total proteins were also subjected to Western blots (**B**, lower panels). Whether the yeast main cultures contained galactose (+) or glucose (-) is indicated. Arabidopsis genes present in the yeast lines are given above the strain number (see

*Figure 2 continued on next page*

*Figure 2 continued*

**Supplementary file 1A** for complete genotypes), and all yeast lines except for the wild type (WT) also contain the *glgC-TM* gene. (**A**) Native PAGE of soluble proteins in glycogen-containing gels that were incubated either with glucose-1-phosphate and phosphorylase (for visualization of branching enzyme and isoamylase activity, top panel) or ADPglucose (for visualization of starch synthase activity, bottom panel). The left- and right-hand panels show separate gels that were processed at the same time. Glucan-modifying activities were revealed by iodine staining. Strains expressing only one branching enzyme (lines T and S) and WT Arabidopsis plant extracts (WS or Col-0 ecotype) are shown for comparison. Enzyme activities in plant extracts, deduced from earlier mutant analyses, are indicated on the right hand side in green (for a representative summary refer to Supplemental Fig. S1 in *Pfister et al. [2014]*). Enzyme activities in yeast extracts, deduced from strain comparisons, are indicated on the left side in black. BE2 migrates as three activities; two very close bands (indicated with the double arrow) and a third slower band that overlaps with BE3 activity (compare strains T and S). SS4 gives a weak but distinct band, as indicated. The branching activity in WT yeast is the endogenous branching enzyme, Glc3p (red asterisk). PHS1, plastidial phosphorylase; PHS2, cytosolic phosphorylase. (**B**) Immunoblots of yeast soluble protein extracts (upper two panels) and yeast total protein extracts (lower two panels), respectively, after separation by SDS-PAGE. The plant sample is the same soluble protein extract in both cases. SS2, GlgC-TM (both carrying a C-terminal HA tag) and SS4 (carrying a C-terminal FLAG tag) were visualized using α-HA or α-FLAG antibodies, as indicated. The expected molecular weights are 83 kDa for SS2-HA, 50 kDa for GlgC-HA and 116 kDa for SS4-FLAG. Soluble and total proteins are shown because each protein's abundance in the soluble extract is influenced by its binding to glucans and the resultant partitioning between the soluble and insoluble fraction. The signal intensities from the soluble proteins should not be compared with those from total proteins as they arise from separate blots.The following figure supplement is available for *Figure 2*:

The following source data and figure supplement are available for figure 2:

**Source data 1.** Glucan contents and $\lambda_{max}$ values of individual replicates.

**Figure supplement 1.** Enzyme function of ISA1/ISA2, BE1, BE2 and BE3 in yeast.

between individual lines are in fact comparable.

We tested the in-vivo functionality of the enzymes by assessing glucan production of yeast strains expressing only single enzyme isoforms. Either plant BE2 or BE3 enabled substantial glucan production in yeast cells lacking their endogenous BE (strains X and Y; *Figure 2—figure supplement 1A*). This was, however, not the case for strains expressing BE1. Here, only low amounts of amylose-like glucans could be detected (strains V and W), comparable to the strain lacking any branching enzyme (strain U; *Figure 2—figure supplement 1A,B*). This suggests that BE1 in fact is not a starch branching enzyme. Since the presence of BE1 also did not alter the glucans made by the other amylopectin biosynthetic enzymes, it was excluded from subsequent strain sets (see *Figure 2—figure supplement 1A,B* and associated text). In contrast, every plant SS produced glucans when expressed as the sole synthase activity (lines 0, 2, 4 and 6; *Figure 3B*). Expression of ISA in wild-type yeast suppressed the accumulation of glycogen (*Figure 2—figure supplement 1D*). This result was consistent with that observed when expressing ISA in *E. coli*, where it also reduced glycogen accumulation (*Sundberg et al., 2013*). Branch-point hydrolysis by ISA may have a degrading effect when it is provided with glycogen as a substrate rather than an amylopectin 'precursor'. The latter presumably crystallizes upon debranching, rendering it inaccessible to further modification (*Sundberg et al., 2013*).

## Yeast strains produce high amounts of glucans

Having established the functionality of the heterologous starch-biosynthetic enzymes, we created a library of over 100 yeast lines. This collection included 30 lines in the mutant background deficient in the endogenous glycogen-metabolic genes (lines 0–29, **Supplementary file 1A**). In these lines, we systematically varied the complement of the four SS isoforms and ISA while keeping GlgC-TM, BE2 and BE3 constant. No loss of heterologous genes could be detected by PCR in line 29 (carrying the highest number of modifications) during strain maintenance and growth in liquid cultures, indicating genetic stability (described in Materials and methods, not shown).

We assessed glucan production in a qualitative manner by growing the 30 yeast lines on plates containing galactose and staining them with iodine vapor (*Figure 3A*). The color and intensity after iodine staining depend on the type of complex formed between iodine and the glucan's secondary structure: Predominantely linear, single-helical glucans such as amylose stain deep blue; double-helical amylopectin chains stain brown, and glycogen (lacking secondary structures) stains weakly red-brown (*Saenger, 1984*; *Manners, 1991*; *Streb et al., 2008*). The engineered yeast lines stained in a

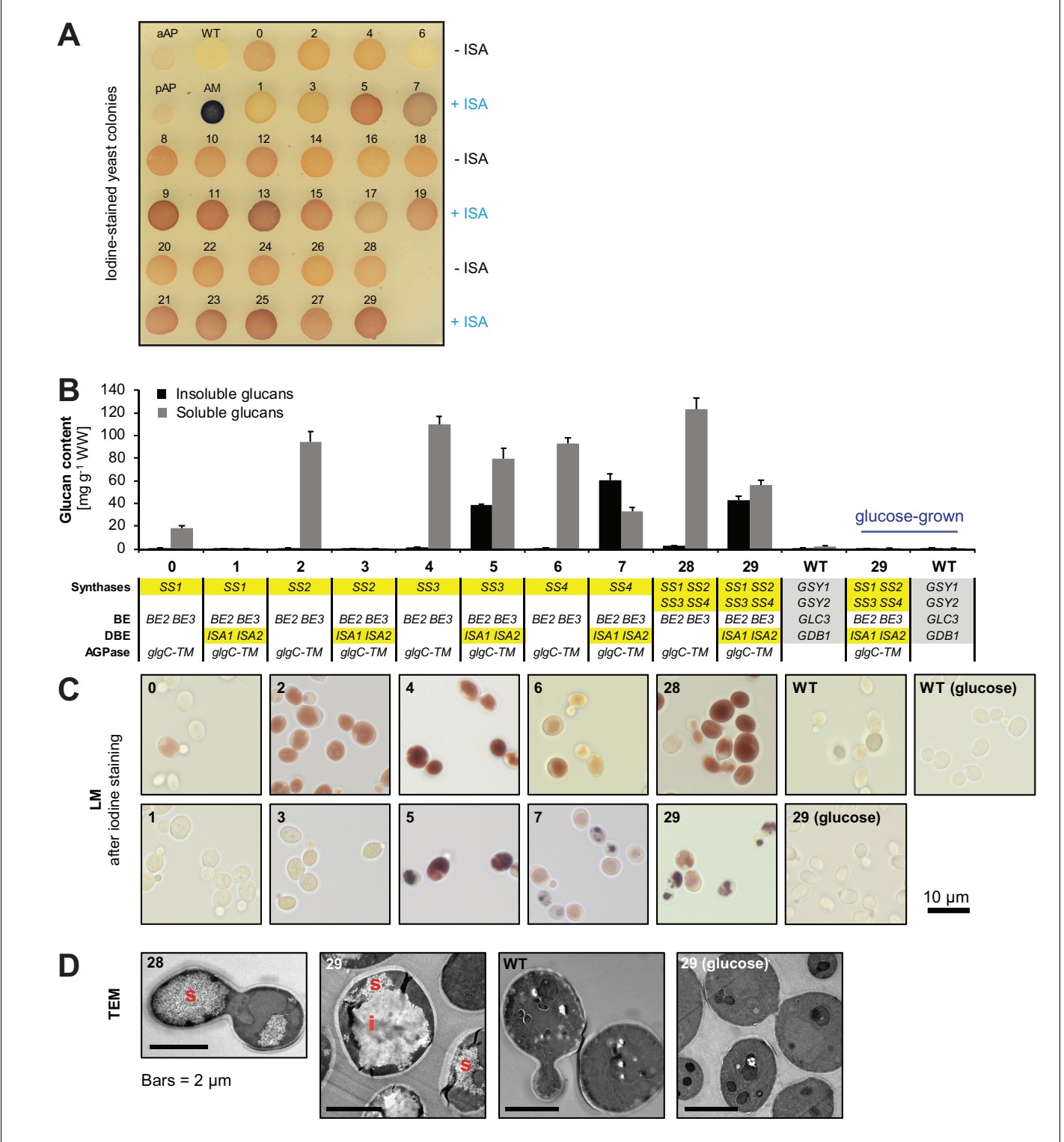

**Figure 3.** Yeast strains synthesize high amounts of glucans. (**A**) Iodine staining of 30 yeast strains (numbered 0–29) which vary the complement of SS isoforms and ISA1/ISA2 in the glycogen-metabolism free background. Cells were grown for 24 hr on plates with galactose and then stained with iodine vapor. Wild-type (WT) yeast, potato amylose (AM) dissolved in DMSO and native, non-gelatinized plant amylopectins from Arabidopsis (aAP) and potato (pAP) are shown for comparison. Indicative genotypes are given in B and *Figure 3—figure supplement 1*. Full genotypes are listed in *Supplementary file 1A*. (**B**) Quantification of insoluble (black bars) and methanol-precipitable soluble (grey bars) glucans of yeast lines grown for 6 hr in liquid main cultures with galactose. The genotypes are indicated; the varying *SS* and *ISA1/ISA2* genes are highlighted in yellow and the endogenous

*Figure 3 continued on next page*

*Figure 3 continued*

yeast genes in grey. For the glucose-grown cultures in **B**-**D**, galactose was replaced by glucose to repress heterologous gene expression. Values are means ± S.D. from 4 replicate cultures (except for the glucose-grown samples, and line 6, where $n$ = 3). WW, wet weight; BE, branching enzyme; DBE, debranching enzyme; AGPase, ADPglucose pyrophosphorylase. (**C**) Light micrographs (LM) from cells of the indicated yeast lines grown as in **B**. Cells were stained with iodine. (**D**) Transmission electron micrographs (TEM) of the indicated yeast lines (grown as in **B**) after chemical fixation. While only particulate, putatively soluble glucans (s) were observed in line 28, line 29 also contained uniform, putatively insoluble glucans (i).

The following source data and figure supplements are available for figure 3:

**Source data 1.** Glucan contents of individual replicates from strains 0 to 29.
**Figure supplement 1.** Quantification of glucans and light micrographs of yeast strains 8 to 29.
**Figure supplement 2.** Dependence of starch synthases on glycogenins.
**Figure supplement 3.** Accumulation of linear malto-oligosaccharides (MOS) in the presence of isoamylase.

variety of shades, suggesting that they produced glucans with different structures. Notably, yeast strains expressing ISA (lines with odd numbers; + ISA) tended to stain darker than their ISA-free counterparts (lines with even numbers; – ISA; placed above).

To quantify glucan accumulation, we cultured yeasts in shake flasks using galactose as the sugar source in the main cultures. The cells were harvested after 6 hr, as growth of yeast strains producing glucans was reduced afterwards (not shown). After homogenization of the cells and centrifugation at 6000 $g$ for 5 min, water-insoluble glucans were collected in the pellet. The supernatant containing soluble glucans was further fractionated by addition of methanol (80% v/v). This yielded a second fraction of water-soluble but methanol-precipitable glucans (i.e. polysaccharides; here named soluble glucans) and a third fraction of methanol-soluble glucans (i.e. short malto-oligosaccharides and free sugars).

Most lines without ISA synthesized high amounts of soluble glucans, which reached around 10% of the wet weight (*Figure 3B* and *Figure 3—figure supplement 1*). Light microscopy (LM) of these cells revealed fairly uniform iodine staining (*Figure 3C* and *Figure 3—figure supplement 1*). Transmission electron micrographs (TEM) from line 28 indicated numerous small particles reminiscent of glycogen (*Figure 3D*). Interestingly, cells expressing just SS1 as a synthase (line 0) accumulated far fewer glucans than lines expressing the other SS isoforms (lines 2, 4 and 6; *Figure 3B*). However, in yeast lines still containing the endogenous glycogenins, all SS isoforms synthesized large amounts of glucan (*Figure 3—figure supplement 2*; see figure legend for details). This suggests that, rather than having low activity, SS1 is less efficient than the other SSs in utilizing available primers or in generating its own.

Most lines expressing ISA produced water-insoluble glucans as well as soluble glucans (*Figure 3B* and *Figure 3—figure supplement 1*). These lines also accumulated linear malto-oligosaccharides, most likely representing chains liberated by ISA (*Figure 3—figure supplement 3*). The presence of insoluble glucans was typically associated with discrete, patchy staining of cells in LM (*Figure 3C* and *Figure 3—figure supplement 1*). This staining showed some variation within single cells and from cell to cell, probably reflecting different stages of yeast cell age and glucan maturation. TEMs from line 29 showed solid continuous glucan particles which often occupied a significant fraction of the yeast cell volume in addition to small, presumably soluble glucan particles (*Figure 3D*).

Interestingly, the lines which expressed ISA but neither SS3 nor SS4 (lines 1, 3 and 9) had very low glucan levels when grown in liquid culture (lines 1 and 3, *Figure 3B*; line 9, *Figure 3—figure supplement 1*; see also lines F, G, J and K, *Figure 3—figure supplement 2*). This again indicates that hydrolysis by ISA may limit glucan accumulation under some conditions (as described above).

## ISA, SSs and BEs have distinct effects on the formation of insoluble glucans

Having a library of strains allows systematic comparisons of glucan accumulation, structure and partitioning between the soluble and insoluble fractions. This enables functional analyses of individual enzymes and an exploration of their interdependencies. Such comparisons furthermore allow us to test whether the specificities evident from plant and in-vitro studies are retained in the yeast.

To monitor the effect of ISA and SSs on the presence of insoluble glucans, we re-calculated these as percentages of total glucans (*Figure 4A*). As mentioned, ISA enabled the synthesis of insoluble glucans (also see *Figure 3B* and *Figure 3—figure supplement 1*). Moreover, we observed a clear effect of SS isoforms on the synthesis of insoluble glucans when ISA was present. Comparing the percentages of insoluble glucans of strains with and without SS4 (e.g., strain 25 *vs.* strain 11) revealed that SS4 consistently promoted the synthesis of insoluble glucans, both in absolute and relative terms (*Figure 4B*). In contrast, SS1 activity rendered the glucans less insoluble and consistently promoted the synthesis of soluble glucans. The presence of SS2 had variable effects, but significantly increased the percentage of insoluble glucans in the absence of SS4 (line 21 *vs.* 11: + 46%, line 15 *vs.* line 5: + 30%). Reciprocally, the increase in the proportion of insoluble glucans caused by SS4 was strongest in the lines without SS2 (line 25 *vs.* 11: + 108%; line 19 *vs.* 5: + 95%).

During the creation of the library of strains described above, we observed lines that produced insoluble glucans despite lacking ISA: Strain I and strain M, containing BE3 as sole BE and either SS2 or SS3, respectively, contained up to 19% of their glucans in an insoluble form (*Figure 4C*). The equivalent strains with both BEs contained only soluble glucans (strains 2 and 4). An effect of BE on the partitioning between soluble and insoluble glucans could also be observed when ISA was present: When together with SS3 and ISA, having BE3 alone promoted insoluble glucan formation (compare strain O *vs.* 5). In contrast, when together with SS4 and ISA, having BE3 alone reduced both the total glucan content and the percentage that was insoluble. In all instances, however, having BE3 alone rather than both BEs, was accompanied by a shift towards longer wavelengths of maximum absorption [$\lambda_{max}$] after iodine staining.

Together, these data show that each enzyme activity influences insoluble glucan formation in a distinct manner. They also highlight the complex functional interplay between both enzyme classes and isoenzymes of each class that are collectively responsible for starch synthesis.

## Chain-elongation specificities of SSs are preserved in the yeast system

Studies of plant mutants and of individual enzymes in in-vitro have suggested that SS classes have distinct preferences in terms of glucan chain elon gation (*Zeeman et al., 2010*; *Tetlow and Emes, 2011*; *Pfister and Zeeman, 2016*). To investigate whether these are preserved in the yeast system, we obtained chain-length distributions (CLDs) from the glucans from strains 0–29. We debranched the glucans, then separated and quantified the resulting linear chains by HPAEC-PAD (high-performance anion-exchange chromatography with pulsed amperometric detection; see *Figure 5A* for examples). We then compared the CLDs from yeast strains either having or not having individual SS isoforms via difference plots.

In all instances, the presence of SS1 increased the relative abundance of shorter chains (degree of polymerization or DP = 6–10) at the expense of intermediate-length ones (DP = 11–25) (*Figure 5B*). This result is in accordance with that from Arabidopsis (*Figure 5B*; *Delvallé et al., 2005*; *Szydlowski et al., 2011*) and cereal mutants (*Fujita et al., 2006*; *McMaugh et al., 2014*). Similarly, the presence of SS2 had a consistent effect on the CLDs of yeast-derived glucans, increasing the relative abundance of intermediate-length chains (*Figure 5C*). Again, these alterations were consistent with previous data of *ss2* mutants from Arabidopsis and other species (*Zhang et al., 2008*; *Edwards et al., 1999*; *Craig et al., 1998*; *Morell et al., 2003*; *Zhang et al., 2004*). The magnitudes of these isoform-specific CLD alterations were smaller among the yeast glucans than among the plant glucans. This is probably due to the overall relative increase in longer chains in the yeast glucans compared to Arabidopsis starch (*Figure 5A*).

SS3 and SS4 did not influence the lengths of short and intermediate chains in a distinct manner (not shown), which is in-line with reports from Arabidopsis mutants (*Zhang et al., 2005*; *Roldán et al., 2007*). However, the tendency of SS3 to increase the abundance of intermediate-length chains in the absence of SS2 (*Zhang et al., 2008*) was also apparent in yeast (*Figure 5D*).

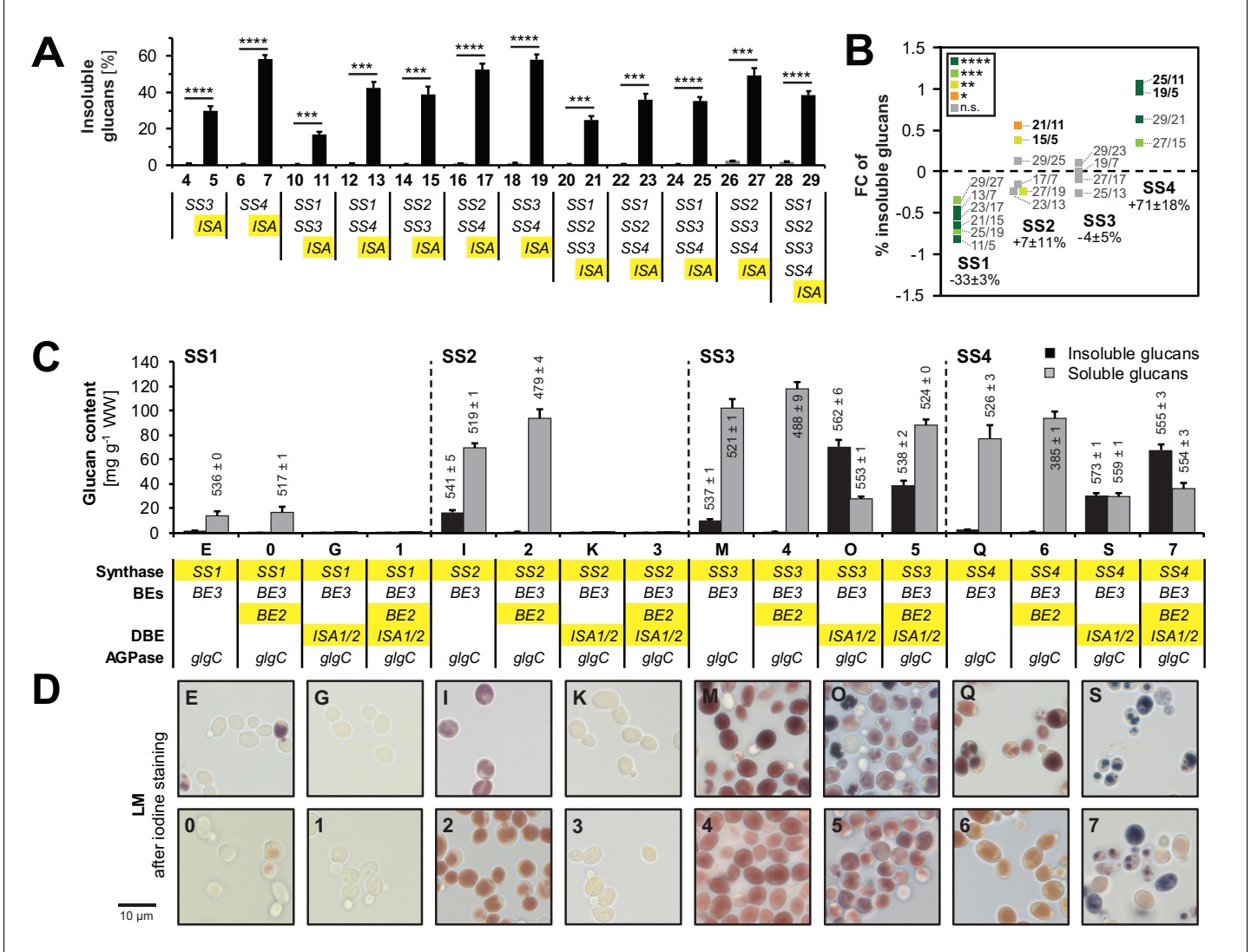

**Figure 4.** Isoamylase, starch synthases and branching enzymes have distinct effects on the formation of insoluble glucans. (A) Percentage of total glucans that were insoluble glucans of lines without isoamylase (grey bars) and their corresponding partners with isoamylase (*ISA*, black bars). Only strain pairs in which both lines have glucan levels > 5 mg g$^{-1}$ wet weight (WW) were included. All lines furthermore contain *BE2*, *BE3* and *glgC-TM*. Values are means ± S.D. from 4 replicate cultures, except for line 6 (*n* = 3). Statistical comparisons were performed using two-sided *t*-tests as described in Materials and methods. The underlying data is available in *Figure 3—source data 1*. ****p value < 0.0001; ***p value < 0.001.(B) Fold changes (FC) of percentage of insoluble glucans depending on individual SS, using data from lines with ISA presented in A. The compared lines are indicated. The comparisons of SS2 in the absence of SS4 and *vice versa* are shown in bold. Given values for each SS are mean percentage changes from all comparisons ± S.E.M. (*n* = 6 for SS1 and SS2, *n* = 4 for SS3 and SS4; see *Figure 4—source data 1* for the calculations). Statistical comparisons were performed using ANOVA as described in Materials and methods. ****p value < 0.0001; ***p value < 0.001; **p value < 0.01; *p value < 0.05; n.s., not significant, p value ≥ 0.05. (C) Quantification of insoluble (black bars) and methanol-precipitable soluble (grey bars) glucans and their wavelengths of maximum absorption after complexion with iodine ($\lambda_{max}$; values above/within the bars, in nm). Yeast lines were grown as described in *Figure 3B*. Values are means ± S.D. from 4 replicate cultures for quantifications and from 2 replicate cultures for the $\lambda_{max}$ measurements (except for insoluble glucans from strains I, 5 and 7 where *n* = 4). The quantification data for lines E, G, I, K, M, O, Q and S are the same as presented in *Figure 3—figure supplement 2*(all yeast strains shown here and in *Figure 3—figure supplement 2* were grown and analyzed together). WW, wet weight; BEs, branching enzymes; DBE, debranching enzyme; AGPase, ADPglucose pyrophosphorylase. (D) Light micrographs of cells from A after staining with iodine. Bar = 10 µm. Source data:

The following source data is available for figure 4:

**Source data 2.** Glucan contents and $\lambda_{max}$ values of individual replicates.

**Source data 1.** Glucan contents of individual replicates from strains 0 to 29.

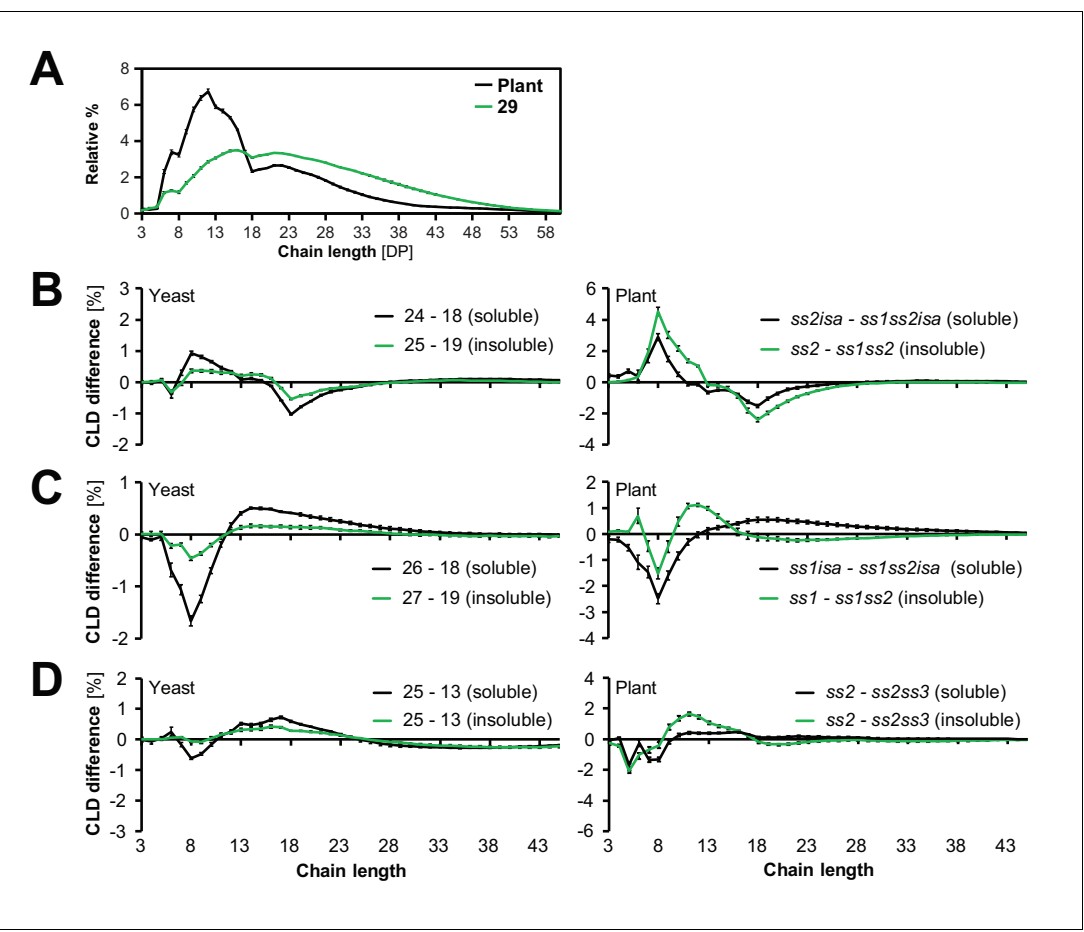

**Figure 5.** Starch synthases retain their chain-elongation specificities. (**A**) Chain length distributions (CLDs) of debranched insoluble glucans from line 29 and of wild-type Arabidopsis starch (WS ecotype). Values are means ± S.D. from 4 replicate yeast cultures or 3 plants, respectively. The CLD of *gbss* mutant Arabidopsis starch is reported to be identical to that from wild-type starch (*Seung et al., 2015*). (**B-D**) Representative graphs illustrating the changes in glucan fine structure upon the addition of SS1 (**B**), SS2 (**C**) or SS3 (**D**) in yeast and *in planta*. Comparisons were done by subtracting the CLDs as indicated (means ± S.E.M., *n* = 4, except for *ss2isa*, *ss1isa*, *ss2*, yeast line 13 (soluble) and *ss2* [insoluble shown in **B** and **D**], where *n* = 3). Data from plants in **B** and **C** are recalculations from *Pfister et al., (2014)*. Horizontal comparisons in yeast and *in planta* involve the same set of amylopectin-synthesizing genes in each case (i.e. line 24 has the whole known complement except for SS2 and ISA1/ISA2, corresponding to an *ss2isa* mutant; line 18 the same enzymes except for SS1, corresponding to an *ss1ss2isa* mutant). Subtracting the CLD of line 18 from that of line 24 thus reveals the effect of the presence of SS1.

The following source data is available for figure 5:

**Source data 1.** Numerical data of chain-length distributions from the plant and yeast glucans.

## Yeast glucans have semi-crystalline properties of starch

The presence of water-insoluble glucans prompted us to test whether these possessed the characteristics of genuine starch granules. We therefore performed a series of microscopic and biophysical analyses, focusing on the insoluble glucans from line 29 (the line expressing all amylopectin-biosynthetic enzymes).

The insoluble glucans were laid down as distinct spherical particles that were several micrometers in diameter, as determined by scanning electron microscopy (*Figure 6A*). The size of these particles was within the range observed for plant starch granules (*Pérez and Bertoft, 2010*), including Arabidopsis leaf starch (*Figure 6A*). Similar, albeit not identical, particles could be purified from other yeast strains producing insoluble glucans, but not from those lacking measurable insoluble glucans

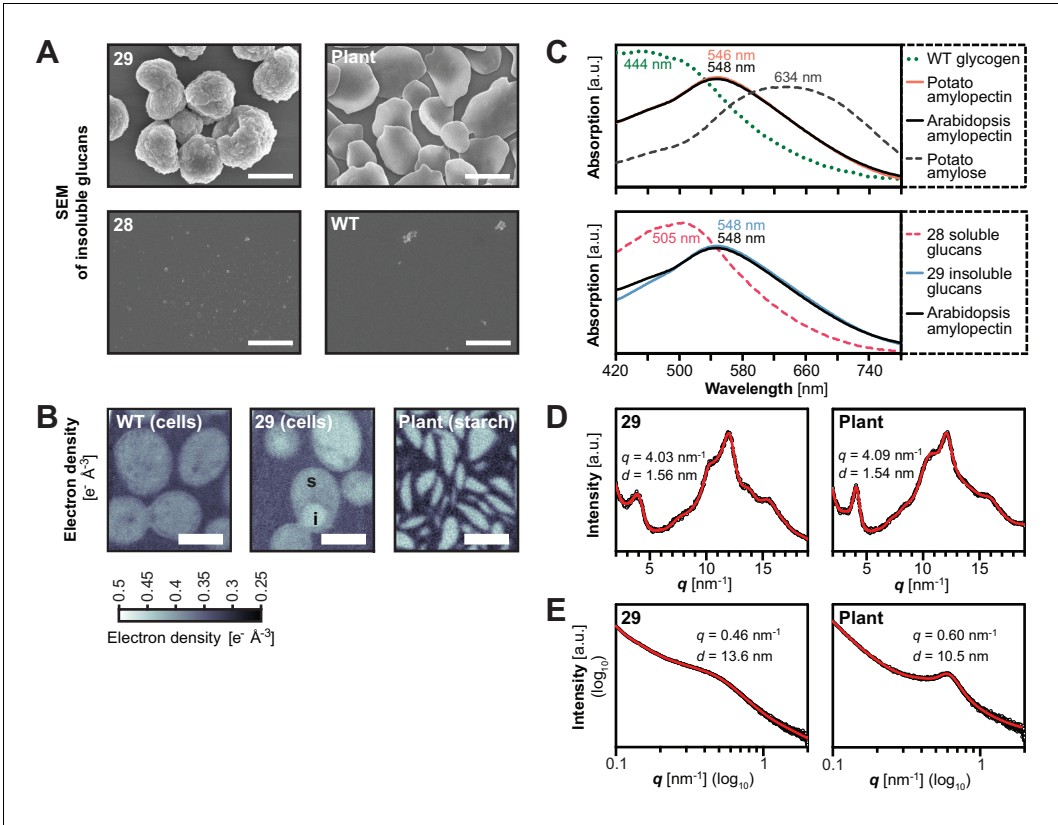

**Figure 6.** Structure of starch granules made in *S.cerevisiae* compared with amylose-free Arabidopsis starch. (**A**) Scanning electron micrographs (SEMs) of purified insoluble particles from line 29 and of amylose-free Arabidopsis starch. No insoluble particles could be purified from WT yeast or line 28. The granules of WT Arabidopsis starch are similar to those of amylose-free starch (*Figure 6—figure supplement 1*). Bars = 2 µm. (**B**) 2D slices through 3D electron density maps obtained by cryo X-ray ptychographic tomography of intact yeast cells (grown as in *Figure 3B*) and purified amylose-free Arabidopsis starch granules. i, putative insoluble glucans; s, putative soluble glucans; bars = 3 µm. (**C**) Absorption spectra of glucans from the indicated yeast lines (WT, line 28 and line 29) and various plant glucans after complexion with iodine. Wavelengths of maximum absorption are indicated. a.u., arbitrary units. (**D**) Wide-angle and (**E**) small-angle X-ray scattering intensity profiles of insoluble glucans from line 29 and amylose-free Arabidopsis starch (i.e. amylopectin). The profiles of WT Arabidopsis starch are similar to those of amylose-free starch (*Figure 6—figure supplement 3*). q, scattering vector; d, lamellar periodicity; a.u., arbitrary units. The following figure supplements are available for *Figure 6*:
The following figure supplements are available for figure 6:

**Figure supplement 1.** Scanning electron micrographs (SEMs) of purified insoluble particles from the indicated yeast strains and from wild-type Arabidopsis (WS ecotype) leaf starch.

**Figure supplement 2.** Absorption spectra of insoluble glucans from the indicated yeast lines.

**Figure supplement 3.** X-ray scattering wild-type Arabidopsis starch.

---

(*Figure 6—figure supplement 1* and *Figure 6A*). The presence of massive insoluble particles is consistent with the observation of dense glucans in TEMs from line 29 (*Figure 3D*).

To investigate the internal glucan structure, we obtained electron density maps of intact yeast cells using cryo X-ray ptychographic tomography (*Dierolf et al., 2010*; *Diaz et al., 2015*). Cells of line 29 contained regions of a high density - putatively insoluble glucans - as well as regions of intermediate density, probably reflecting soluble glucans. Remarkably, the high-density regions had a density identical to that of Arabidopsis amylopectin, pointing towards a comparable internal

organization of the glucans (mean electron density [± S.D., $n$ = 6] from line 29 = 0.44 ± 0.01 e$^-$Å$^{-3}$, corresponding to a mass density of 1.36 ± 0.04 g mL$^{-1}$; Arabidopsis amylopectin density = 0.441 ± 0.004 e$^-$ Å$^{-3}$, corresponding to 1.36 ± 0.01 g mL$^{-1}$; see Materials and methods for details) (*Figure 6B*).

Further, the absorption characteristics of the insoluble glucans after iodine complexion from line 29 were almost identical to those of amylopectins from wild-type and *gbss* mutant plants ($\lambda_{max}$ after iodine-complexion of ~548 nm; *Figure 6C*; *[Zeeman et al., 2002b]*). In addition, they were distinct from that of highly-branched, soluble glycogen from wild-type yeast ($\lambda_{max}$445 nm), from that of soluble glucans from the corresponding line without ISA (line 28, $\lambda_{max}$505 nm) and from that of linear glucans such as potato amylose ($\lambda_{max}$634 nm). The similarity in absorption spectra indicates that the insoluble glucans from line 29 form comparable secondary structures to those in plant amylopectin, i.e. double helices between adjacent chains. Notably, also the iodine absorption spectra of insoluble glucans from yeast strains expressing only subsets of SSs (e.g. SS3, line 5, or SS4, line 7, or SS1, SS2 and SS3, line 21) were dissimilar to that from line 29, re-affirming the influence of SSs on glucan structure (*Figure 6—figure supplement 2*).

To directly investigate the formation of secondary and tertiary structures in the insoluble glucans from line 29, we acquired X-ray scattering intensity profiles. Wide-angle X-ray scattering (WAXS) of purified and dried glucans resulted in a diffraction pattern typical of many starches, where glucan chains form double helices that align with an inter-helical spacing of ~1.6 nm, indicative of a B-type allomorph (*Kong et al., 2014*; *Figure 6D*). Small-angle X-ray scattering (SAXS) indicated that the helices were stacked in a weakly ordered lamellar structure with a periodicity of 13.6 nm (*Figure 6E*), slightly longer than typically reported for plant starches (*Pérez and Bertoft, 2010*). Here, Arabidopsis starch had a ~10.5 nm periodicity (*Figure 6E* and *Figure 6—figure supplement 3*). Chain-length distribution (CLD) analyses revealed that the insoluble glucans had relatively more long chains (DP > 18) than wild-type Arabidopsis amylopectin. Nevertheless, the key characteristics of amylopectin were preserved as they grouped in sub-populations of chains with DPs of 6–8, 9–18 and >18 (*Figure 5A*).

Collectively, these analyses show that the insoluble glucans from line 29 form granular particles which display the typical internal organization of semi-crystalline plant starch.

## Discussion

To our knowledge, this study represents the first example of where insoluble, starch-like granules have been produced in a non-plant system. It provides direct evidence that SSs, BEs and ISA are all key enzymatic components for starch granule formation. We furthermore show that the heterologously expressed enzymes retain their specificities in yeast and propose that this system can be used as an advanced tool for studying the overall process of starch biosynthesis in unprecedented speed and depth.

Despite decades of work with plant mutants, the requirements for starch synthesis have not been clearly defined. A major obstacle associated with any plant-related approach is the presence of additional starch-active enzymes, some of which may not even be known. In Arabidopsis chloroplasts, where starch is made and degraded during a single diurnal cycle, there is a plethora of starch-catabolising enzymes: the two additional debranching enzymes ISA3 (At4g09020) and limit-dextrinase (At5g04360), at least two active β-amylases (BAM1 [At3g23920] and BAM3 [At4g17090]), the α-amylase AMY3 (At1g69830), the disproportionating enzyme DPE1 (At5g64860) and the α-glucan phosphorylase PHS1 (At3g29320) (*Zeeman et al., 2004*, *2007*; *Delatte et al., 2006*; *Fulton et al., 2008*; *Seung et al., 2013*). In general, starch synthesis and degradation are regarded as separate processes in wild-type plants. It is possible that the rapid crystallization of correctly synthesized amylopectin protects it from further modification by enzymes of starch degradation. However, in Arabidopsis mutants deficient in multiple starch-biosynthetic enzymes, glucan synthesis and accumulation have been repeatedly shown to be influenced by degrading enzymes, for example in the *isa1isa2* (*Delatte et al., 2005*), the *be2be3* (*Dumez et al., 2006*), the *isa1isa2isa3lda* (*Streb et al., 2008-*), the *ss2ss3* (*Pfister et al., 2014*) and the *ss3ss4* mutant (*Seung et al., 2016*). Consequently, it has been very difficult to reliably ascribe the glucans produced in a plant's cell to the actions of a distinct enzymatic set.

The approach we used here to deduce the essential components for starch biosynthesis is more direct. Although plant genes have been expressed in yeast or *E. coli* before (*Guan et al., 1995*; *Guan and Keeling, 1998*; *Seo et al., 2002*), our study is unique in several ways. The yeast is purged of its endogenous glycogen-metabolic pathway and the introduced genes can be induced together. These factors allow us to study starch biosynthesis independently of interfering background effects from the yeast itself, or from the plant's degradative enzymes. Our study is also unique in terms of the systematic variation in enzyme combinations. This enables the contribution of each enzyme and capacities of different enzymatic sets to be analyzed in ways that were previously not possible.

## Role of ISA, SS isoforms and BEs during the formation of insoluble glucans

Our work confirms that debranching by ISA is the major driving force for the formation of insoluble glucans: In most (but not all) cases, insoluble glucans were only obtained when ISA was present (*Figure 4A*). The promotion of insoluble glucans by ISA is in-line with plant mutants lacking this enzyme, which typically accumulate water-soluble polysaccharides, so-called phytoglycogen (*Nakamura et al., 1996*; *Mouille et al., 1996*; *Zeeman et al., 1998*; *Dinges et al., 2001*; *Burton et al., 2002*; *Bustos et al., 2004*). Presumably, ISA selectively hydrolyzes excessive or mis-placed branches that would otherwise interfere with the formation of double helices (*Myers et al., 2000*). Consistent with this, we could detect a pool of linear malto-oligosaccharides in strains where ISA promoted insoluble glucan formation (*Figure 3—figure supplement 3*). In wild-type plants, such a malto-oligosaccharide pool is probably rapidly metabolized and therefore not detectable (*Streb et al., 2008*).

In the present study, we also dissected the contribution of SSs and BEs to insoluble glucan formation. When ISA was present, SS1 always decreased the production of insoluble glucans (*Figure 4B*). The decreased crystallinity of glucans in the presence of SS1 probably stems from the increased abundance of glucan chains of DP 6–10 (*Figure 5B*) that are too short to efficiently contribute to the formation of higher-order structures underlying starch crystallinity (*Pfannemüller, 1987*; *Gidley and Bulpin, 1987*). We have recently described a comparable effect of SS1 in Arabidopsis, where the lack of SS1 in the *isa* mutant background restored granule formation in the mesophyll cells (*Pfister et al., 2014*). In contrast, SS4 enhanced the production of insoluble glucans. This effect cannot be traced back to distinct structural modifications – no consistent alterations in glucan structure by SS4 were observed in our comparisons or in studies of Arabidopsis *ss4* mutants (data not shown; *Roldán et al., 2007*; *Szydlowski et al., 2009*). One has to bear in mind, though, that the methods currently available for the analysis of starch structure give an incomplete picture, and structural features introduced by SS4 may not have been revealed.

We did not observe uniform effects upon the expression of SS2 and SS3 (*Figure 4B*). For SS2, this was surprising; it has been shown that SS2 specifically elongates chains from around DP 8 to around DP 13 and suggested that this promotes the formation of higher-order structures (*Nakamura et al., 2005*; *Fujita et al., 2012*; *Pfister et al., 2014*). Indeed, mutation of *SS2* in Arabidopsis results in small amounts of phytoglycogen and enhances the accumulation of phytoglycogen in *isa* mutants of both rice and Arabidopsis (*Fujita et al., 2012*; *Pfister et al., 2014*). Although the alterations in glucan structure introduced by SS2 are preserved in the yeast system (*Figure 5C*), SS2 promoted the synthesis of insoluble glucans only when SS4 was absent (*Figure 4B*, comparisons in bold). It is possible that the strong effect of SS4 in promoting insoluble glucan formation masks this effect of SS2 in yeast, potentially due to differences in relative expression levels compared to Arabidopsis.

Our data also add new insights into how branching level influences the synthesis of insoluble glucans: In some lines expressing only BE3 as a branching enzyme, insoluble glucans were obtained despite the absence of any debranching activity (*Figure 4C*), confirming that the latter is not an absolute requirement for the insoluble-glucan production (*Streb et al., 2008*). BE2 and BE3 are closely related class II BEs and largely redundant in Arabidopsis (*Dumez et al., 2006*). Therefore, the impact of having only BE3 is probably due to overall branching activity rather than to a difference in specificity. It is striking that in combination with some, but not all SS this low BE activity resulted in formation of insoluble glucans. This was despite the fact that in all cases the $\lambda_{max}$ of the iodine-stained glucans increased, suggesting that the structure was altered in a similar way, i.e. more chains long enough to form single or double helices were present. Clearly, such secondary structures are

not sufficient for insoluble glucan formation, presumably because they need also to align in parallel to form a semi-crystalline matrix. Further investigations of the insoluble glucans made by different strains in our yeast collection will deepen our understanding of the structural features that allow crystallization.

## Seeding of glucan synthesis by SS isoforms

Glycogen biosynthesis in yeast and animals begins with the self-glycosylation of glycogenins on tyrosine residues, yielding short glucan chains that are then elongated by glycogen synthases (*Rodriguez and Whelan, 1985*; *Lomako et al., 1988*; *Pitcher et al., 1988*; *Cheng et al., 1995*). In contrast, the mechanism of starch initiation is not known. In Arabidopsis, genetic evidence suggests a specialized role for SS4 in this process since *ss4* mutants display a strong reduction in starch granule number (*Roldán et al., 2007*). Although *ss3* mutants have normal number of starch granules (*Seung et al., 2016*), further loss of SS3 from the *ss4* mutant background resulted in a nearly complete failure of glucan synthesis (*Szydlowski et al., 2009*), indicating that SS3 becomes indispensable for glucan priming when SS4 is absent.

In our yeast strains, it is unclear what primes glucan synthesis. Every SS was capable of glucan synthesis on its own (*Figure 3B*), but to varying extents. Particularly, when SS1 was the sole synthase, the overall glucan content of the whole population was low and iodine staining revealed that only a fraction of the cells accumulated glucans (strains 0 and E, *Figure 3C,D* and *Figure 3—figure supplement 2*). However, glucan synthesis by SS1 was strongly improved in a corresponding strain with the endogenous glycogenins (*GLG1* and *GLG2*; strain D, *Figure 3—figure supplement 2*). This phenotype is reminiscent of the yeast glycogenin mutant (*glg1glg2*) which synthesizes only little glycogen overall but accumulates it in single colonies, presumably due to stochastic initiation events (*Torija et al., 2005*). This suggests that, similar to yeast glycogen synthases, SS1 has limited capacity to initiate glucan synthesis, but can use primers provided by glycogenins. It is worth noting that SS2 also synthesized slightly fewer glucans when glycogenins were absent (strain H *vs.* I, *Figure 3—figure supplement 2*).

Our study also suggests that degradation influences glucan initiation. When only SS1 and/or SS2 were present, glucan accumulation was strongly diminished by ISA (e.g., in strains 1, 3 and 9; *Figure 3* and *Figure 3—figure supplement 1*; also see *Figure 3—figure supplement 2*). We first reasoned that SS1 and SS2 alone may generate a polymer that is excessively debranched by ISA, possibly because the glucan fails to become crystallization-competent upon debranching and hence remains accessible to further ISA action. However, the absence of malto-oligosaccharides - the expected products of ISA action - in these lines (*Figure 3—figure supplement 3*) argues against such a scenario. Rather, the degradation may happen at a very early time point and could impair the glucan synthesis altogether by limiting the availability of primers. This observation is particularly interesting since Arabidopsis *ss3ss4* mutants are virtually glucan-free due to an apparent failure in the priming of glucan synthesis (*Szydlowski et al., 2009*). Indeed, it was recently shown that the additional knock-out of the α-amylase AMY3 in *ss4* Arabidopsis mutants increases granule number and overall starch content, and loss of AMY3 in *ss3ss4* Arabidopsis mutants (*amy3ss3ss4* mutants) even restored some granule formation (*Seung et al., 2016*). These phenotypes were interpreted to mean that at least part of the priming function of SS4 lies in the protection of primers from premature degradation. Our observation that ISA may also diminish primers could explain why glucan accumulation in *amy3ss3ss4* mutants is only partially restored. Notably, ISA has been implicated as a negative regulator of granule initiation before, since the remaining starch of *isa* mutants is typically deposited as numerous small granules (*Burton et al., 2002*; *Bustos et al., 2004*). The generation and analysis of plant mutants deficient in SS3, SS4 and different starch degrading enzymes is ongoing in order to test this hypothesis.

## Differences between the yeast glucans and plant starch

It is interesting - but perhaps not surprising - that the granules made in yeast by the Arabidopsis enzymes are not identical to those made by Arabidopsis itself. For instance, long chains (DP > 18) were overrepresented in the yeast glucans, in particular when ISA was present (*Figure 5A* and data not shown). In plant amylopectin, the group of chains with DP < 25 is believed to make up a single 9- to 10.5-nm-repeat, being part of the double helices that underlie starch crystallinity (so-called A

and B$_1$ chains; *Hizukuri [1986]*). Since the length of these chains probably defines the width of the lamellar repeat, the relative abundance of long chains in yeast glucans may explain the increased width of the repeat we observed (13.6 nm in the insoluble glucans from line 29; *Figure 6E*).

The molecular factors determining the width of the lamellar repeat are not known, but were suggested to involve the specific placement of branches by BEs (*Nakamura, 2002*). Nonetheless, we did not obtain a wild-type repeat distance, despite having expressed both known starch BEs from Arabidopsis. Most plants contain an additional BE of class I, which displays distinct specificities when assessed in vitro (*Rydberg et al., 2001*; *Nakamura et al., 2010*; *Sawada et al., 2014*), but the Arabidopsis' genome does not encode such a BE (*Dumez et al., 2006*). In-line with earlier reports (*Dumez et al., 2006*; *Wang et al., 2010*), our data also do not support the idea that the gene annotated as BE1 in Arabidopsis represents a third BE activity (*Figure 2—figure supplement 1*).

While we believe that we have expressed the right enzyme combinations, differences in the relative amounts of each enzyme between the yeast lines and Arabidopsis are likely. Indeed, although the BE activities according to our zymograms appear comparable in yeast and in Arabidopsis leaves, some SS activities differ (*Figure 2A*). In particular, SS3 activity was elevated in yeast compared with Arabidopsis. The same may be true for SS4, which was proposed to contribute only little to chain elongation in Arabidopsis (*Szydlowski et al., 2009*). It is interesting to note that the repression or mutation of class II BEs causes the so-called *amylose-extender* phenotype, where amylopectin is characterized by longer chains (*Boyer et al., 1980*; *Hedman and Boyer, 1982*; *Mizuno et al., 1993*; *Sestili et al., 2010*; *Regina et al., 2010*; *Klucinec and Thompson, 2002*). In barley, this was shown to be accompanied by a longer lamellar repeat (12.5 nm instead of 10.4 nm; *Regina et al., [2012]*). However, it is unclear whether this phenotype reflects a limitation of overall BE activity (creating an imbalance with chain elongation) or the specificity of the remaining class I BE, or both. In a recent molecular genetic approach, an impact on amylopectin structure either by BE specificity or by the amount of BE activity relative to the other starch-biosynthetic enzymes was shown directly. *Lu et al., (2015)* complemented the *be2be3* Arabidopsis mutant with class I or class II BEs. Analysis of different transgenic lines established that when either BE activity was limiting, it resulted in decreased amounts of starch and amylopectin with a higher proportion of chains of DP > 18, compared with when the same BE was expressed at higher levels. It would be interesting to see if, in such transgenic Arabidopsis, there is variation in the length of the lamellar repeat.

Another difference between the glucans made in yeast and in plant cells concerns the presence of soluble polysaccharides: while virtually all plant glucans are normally laid down as insoluble starch granules, every yeast strain accumulated also substantial amounts of soluble glucans. The percentages of insoluble glucans from total glucans varied depending on the complement of enzymes present (as discussed above). Still, it did not exceed ~70% in the best cases (strains O and N, *Figure 4C* and *Figure 3—figure supplement 2*) and ranged between 17 and 58% when ISA and both BEs were present (*Figure 4A*). This incomplete formation of insoluble glucans in yeast can have several reasons. First, it could again be a matter of enzyme balance, although we do not know of a comparable instance from plants here. Second, the efficient crystallization of glucans may require additional factors that we have not yet introduced into yeast. An interesting candidate is Early Starvation1 (or ESV1), a non-catalytic starch-binding protein whose absence causes altered granule morphology and starch turnover in Arabidopsis (*Feike et al., 2016*). ESV1 was proposed to facilitate the correct alignment of glucans within the granule matrix, which in turn could indirectly control the accessibility and thus degradation rate of glucans. However, its precise action remains unclear. Third, it is possible that soluble glucans that fail to crystallize efficiently are also synthesized in plant cells, but are rapidly removed by the starch-degrading enzymes described above. In yeast, such soluble glucans would accumulate as soluble glucans, since the degrading machinery is absent. Finally, it is also important to note that our definition of insoluble and soluble glucans in this context is based on the fractionation by centrifugation at 6000 *g* for 5 min. Insoluble glucans with a tiny particle size may remain in the supernatant after such a low-speed centrifugation. Indeed, preliminary experiments suggest that half of the glucans in the soluble fraction from line 29 sediments upon extending the centrifugation to 20 min (data not shown), indicating that at least part of it may actually be insoluble.

## Future applications of the yeast system

Further work should allow us to dissect the causes of the structural differences described above and simultaneously take us closer to a holistic view of starch biosynthesis. For instance, by applying

quantitative proteomics, we could assess the relative differences in enzyme levels between yeast and Arabidopsis. By exerting further control over the relative expression level of the starch-biosynthetic enzymes in yeast, we would expect to obtain insoluble glucans with an even closer match to genuine Arabidopsis leaf starch. Furthermore, the systematic variation of the relative activities could allow as to determine the impact of enzyme level as well as enzyme specificity. This could be achieved using well-defined modules for heterologous gene expression (*Lee et al., 2015*), benefitting from the recent large-scale characterization of yeast promoters, 5'UTRs and terminator sequences (*Keren et al., 2013*; *Dvir et al., 2013*; *Yamanishi et al., 2013*).

Our system may also be optimal for exploring the special characteristics of each enzyme in detail by studying the effects of random or site-directed mutations, domain swaps etc. Further, it provides exciting new opportunities to explore other important, but less-well understood aspects of starch granule synthesis. For instance, a number of starch-biosynthetic enzymes, have been found to be phosphorylated (*Tetlow et al., 2004*, *2008*; *Grimaud et al., 2008*; *Makhmoudova et al., 2014*; *Kötting et al., 2010*) and, in case of cereals, to assemble into protein complexes (*Tetlow et al., 2008*; *Hennen-Bierwagen et al., 2008*, *2009*; *Liu et al., 2009*; *Crofts et al., 2015*; *Ahmed et al., 2015*). Still, whether and how these protein modifications and associations affect the enzyme activities has mostly remained elusive. In yeast, these questions could be readily addressed by expressing putative regulatory kinase/phosphatase pairs or enzymes with mutated phosphorylation sites or interaction domains.

Our understanding of industrially important starch traits, such as starch phosphorylation or amylose content could also be greatly improved. Enzymes controlling starch phosphorylation (i.e., glucan, water dikinases and glucan phosphatases; *Silver et al., [2014]*) could be expressed and studied in yeast, as could factors affecting amylose content. For the latter, one could integrate GBSS with its recently-identified granule-targeting protein PTST (Protein Targeting to Starch, *Seung et al., 2015*), with or without enzymes altering the levels of malto-oligosaccharides that may serve as primers of amylose synthesis (e.g., starch-degrading enzymes; *Zeeman et al., 2002a*). There is also the potential to study non-enzymatic proteins, such as ESV1 (*Feike, 2016*) or the rice protein Floury Endosperm6 (FLO6; *Peng et al., 2014*). These proteins are of particular interest since they appear to exert their function through binding to starch itself or to starch-biosynthetic enzymes, thus providing an additional level of enzyme regulation. Furthermore, our system could be used to investigate the potential of non-plant enzymes, for instance 4,6-$\alpha$-glucanotransferases (*Bai et al., 2015*), to modify starch polymers and thereby improve their properties. Such functional aspects could also be explored given that yeast culture is scalable.

Having illustrated the feasibility of this approach using the Arabidopsis genes, we suggest that it could easily be applied to study starch-biosynthetic pathways in crop species where knock-out mutants are rarer and/or more tedious to create (due to generation times, polyploidy, etc.). Even with advanced genome editing and transgenesis methods, the creation of a set of homozygous plant lines may take substantially longer than recreating and analyzing the pathway in detail in yeast, particularly in the light of the emerging systems for combinatorial gene assembly (*Weber et al., 2011*; *Lee et al., 2015*). The yeast platform may thus be used to perform comprehensive pioneering studies to identify promising enzyme combinations for subsequent testing in plants. Overall, this system could usher in a new era in starch biosynthesis research and inspire work ranging from theoretical modelling of the biosynthetic process to the strategic analysis of functional properties of novel glucans for industrial use.

## Materials and methods

### Chemicals, media and plant materials

Unless otherwise noted, chemicals were purchased from Sigma-Aldrich and restriction enzymes from Fermentas (Thermo Fisher Scientific).

Complex medium consisted of 1% (w/v) Bacto yeast extract (BD) and 2% (w/v) Bacto peptone (BD), supplemented with either 2% (w/v) glucose (medium named YPD) or 2% (w/v) galactose (Acros Organics; medium named YPGal). Sugars were added after autoclaving; 20% (w/v) glucose stock was separately autoclaved and 20% (w/v) galactose stock was filter-sterilized through a 0.22 µm filter unit. Minimal medium lacking nitrogen (MIN-N medium) was prepared as described

(*Johnston et al., 1977*), except for the replacement of $FeCl3_3 \times 6\ H_2O$ (10 µg $L^{-1}$) by $NH_4Fe(SO_4)_2 \times 10\ H_2O$ (16.5 µg $L^{-1}$) and the addition of histidine and uracil (20 mg $L^{-1}$each). The medium was supplemented with either 2% (w/v) glucose (named MIN-N-Glu) or 2% (w/v) galactose (Acros Organics; medium named MIN-N-Gal). YPD plates were prepared as for liquid medium, but included 2% (w/v) bacto agar (BD). Plates containing synthetic complete (SC) or complex medium with glycerol (YPG) were prepared as described (*Sherman et al., 1986*), except for a higher concentration of leucine in SC media (60 mg $L^{-1}$). Uracil was omitted in SC-ura plates. Plates with synthetic medium and galactose (SCgal), used for iodine staining of yeast colonies were prepared like SC plates but with replacement of glucose by 2% (w/v) galactose. Plates with 5-fluoroorotic acid (FOA plates) are SC plates supplemented with 0.1% (w/v) 5-fluoroorotic acid (Fermentas).

The underlying alleles of the homozygous *Arabidopsis thaliana* mutants are listed in *Supplementary file 1E*. Wild-type (WT) Arabidopsis (either of Wassilewskija [WS] or Columbia-0 [Col-0] ecotype) and mutant plants were grown as described (*Streb et al., 2008*). Unless otherwise noted, plants were grown for four weeks and harvested at the end of the light period for maximum leaf starch content. Purified WT Arabidopsis leaf starch from Col-0 ecotype (used for SEM) and amylose-free Arabidopsis leaf starch from *gbss ptst* mutants (*Seung et al., 2015*) (used for X-ray scattering, X-ray tomography, iodine staining and iodine absorption spectra) were kindly donated by David Seung. Potato amylopectin is tuber starch isolated from the *amf* amylose-free variety (*Visser et al., 1991*) (kindly provided by Prof. Richard Visser) and potato amylose is type III from Sigma.

## Generation of yeast strains

CEN.PK113-11C (*MATa MAL2-8C SUC2 his3Δ ura3-52*) was kindly provided by Prof. Barbara A. Halkier, Department of Plant and Environmental Sciences, University of Copenhagen, Denmark. Yeast strains were generated by sequential integration of constructs either at loci of the yeast expression platform (*Mikkelsen et al., 2012*) (designed for stable multiple gene expression) or at loci of the glycogen-metabolic pathway (described below). The genotypes of *S. cerevisiae* strains used in the present study are presented in *Supplementary file 1A*.

### Cloning of constructs and generation of yeast integration vectors

Coding sequences (CDS) of Arabidopsis genes less the chloroplast transit peptides as predicted by ChloroP (*Emanuelsson et al., 1999*) were cloned by PCR. In the case of BE2, a shorter sequence than predicted was removed (48 instead of 61 amino acids) since a peptide starting from amino acid 49 has been detected in leaf proteomics (pep2pro database [*Baerenfaller et al., 2011*]). In addition to the ATG start codon, a glycine or alanine codon was introduced immediately downstream to improve the Kozak sequence (*Hamilton et al., 1987*), if necessary (*Supplementary file 1C*). A sequence codon-optimized for *S. cerevisiae* (BIOMATIK) was used for *SS2* as it was not expressed otherwise. The *E. coli* APGase *glgC-TM* carries three amino-acid changes (R67K, P295D, G336D) that render it insensitive to allosteric regulation (*Sakulsingharoj et al., 2004*). *SS2*, *SS4* and *BE1*, the activity of which is not detectable in plant extracts, were C-terminally fused to *HA* (for SS2) or *FLAG* tags (for SS4 and BE1), respectively. A C-terminal *HA* tag was also added to *glgC-TM*. All CDS were fused to a ca. 450 bp $P_{GAL1}$ promoter, except for ISA1 and ISA2 that were fused to the bidirectional $P_{GAL10}$-$P_{GAL1}$ promoter (*Figure 1—figure supplement 1*). These promoter fusions were cloned into yeast integration vectors pX-2, pX-4, pXI-2, pXII-1, pXII-2 and pXII-5 of the yeast expression platform (*Mikkelsen et al., 2012*) or modified variants (described below) using USER fusion (*Geu-Flores et al., 2007*).

To target constructs to other loci, the two flanking recombination sites (5' site named 'up' and 3' site named 'down') were replaced by other CEN.PK113-11C sequences of between 300 and 800 bp in length. First, a 'GSY1 down' construct, targeting the 3'end of the *GSY1* gene, was PCR-amplified from chromosomal DNA of CEN.PK113-11C using primers with 5' *Sbf*I and 3' *Asc*I and *Bcl*I restriction sites (*Figure 1—figure supplement 1C*). The product was then cloned into pJET1.2 (Life technologies), cut with *Sbf*I and *Bcl*I and ligated into a pX-3 vector digested with the same restriction enzymes, resulting in the vector pX-3_GSY1_down. Similarly, a 'GSY1 up' construct, targeting the 5' UTR and beginning of the *GSY1* coding sequence, was PCR amplified from chromosomal DNA with primers carrying a 5' *Avr*II restriction site and 3' *Sac*II and *Bst*API sites, cloned into pJET1.2, cut with *Bgl*II (a close site on pJET1.2) and *Bst*API and ligated into the pX-3_GSY1_down vector digested

with the same restriction enzymes. The resulting vector was named pGSY1. Vectors targeting *GSY2*, *GLC3*, *GLG1*, *GLG2*, *GPH1*, *GDB1* were generated by replacing the recombination sites of pGSY1 by the sequences homologous to the beginnings and ends of the respective genes using the 5' *Avr*II and 3' *Sac*II restriction sites (for the recombination sites 'up') or the 5' *Sbf*I and 3' *Asc*I sites (for the recombination sites 'down'), respectively. All restriction sites were added via primers during PCR amplification on CEN.PK113-11C chromosomal DNA. The resulting vectors were either used directly for disruption of genes (pGLG1, pGLG2, pGPH1, pGDB1) or modified to contain a starch-biosynthetic gene as described above (pGSY1, pGSY2, pGLC3). Plasmids were verified by sequencing. Details on plasmids and primers for their generation are provided in *Supplementary file 1B, C* and *Figure 1—figure supplement 1C*.

## Transformation, confirmation of insertion and test of genetic stability of yeast

Yeast transformation, selection on SC-ura plates and strain purification were performed as described (*Mikkelsen et al., 2012*). Insertion of constructs at the expected loci was confirmed by multiple PCRs targeting upstream and downstream integration sites (*Supplementary file 1D*). The correct insertion was further confirmed by sequencing the insertion sites in >99% of the insertions. To recycle the *URA3* marker for subsequent transformation, yeasts were selected on plates containing 0.1% 5-fluoroorotic acid (FOA plates), positive colonies streak purified on YPD and purified colonies tested for petiteness on YPG plates and loss of marker on SC-ura plates. Marker loss was further confirmed by PCR (*Supplementary file 1D*). Final lines were additionally tested for the presence of all expression constructs by PCR that spanned promoter and coding sequences of the expression constructs (depicted in the last eight rows of *Supplementary file 1D*).

The PCRs spanning promoter and coding sequences were also employed for testing the genetic stability during strain maintenance and growth in line 29, which carried the highest number of modifications. For this, cells from a glycerol stock were first cultivated on an YPD plate and then grown in liquid cultures with complex media (preculture with glucose, main culture with galactose, i.e., inducing condition, as described below). After incubation of the main culture for 6 hr, single-cell colonies were obtained by plating cells on an YPD plate. No loss of construct loss was detected among all 42 single-cell colonies tested by stringent PCR spanning promoter and coding sequences of the expression constructs (depicted in the last eight rows of *Supplementary file 1D*, data not shown).

## Growth of yeasts in shake-flask experiments

Yeasts were grown at 30°C with 260 rpm shaking. Unless otherwise specified, all yeasts were grown in complex medium. Yeast lines from YPD plates were first inoculated in YPD and grown overnight. Cells from these pre-cultures were inoculated in flask main cultures with either YPGal (medium with galactose; inducing condition) to a starting optical density at 600 nm ($OD_{600}$) of 0.3 or YPD (medium with glucose; repressing condition) to a starting $OD_{600}$ of 0.1. After growing for 6 hr, cells were pelleted, washed twice with water and the cell pellet flash frozen in liquid nitrogen. The wet weight is the weight of the cell pellet after carefully removing the supernatant prior to freezing. Samples were stored at −80°C. Replicate cultures arise from independent pre-cultures of a single yeast line. Yeasts were grouped according to replicate number (instead of line/genotype) in a non-random way during growth and processing.

For growth of yeast lines that express BE3 or ISA1/ISA2 only (*Figure 2—figure supplement 1D*), a specific protocol was used to induce the glycogen biosynthesis pathway (e.g., glycogen synthases). Yeasts from YPD plates were first inoculated in YPD for overnight cultures, then inoculated in YPgal (inducing condition) or YPD (repressing condition) to a starting $OD_{600}$ of 0.6 and grown for 6 hr. To promote glycogen biosynthesis, yeasts were transferred to minimal medium lacking nitrogen (MIN-N-gal or MIN-N-glu medium, respectively). Therefore, the cells were pelleted, once washed with the respective MIN-N medium and then resuspended in the MIN-N medium to reach the same volume as in YPgal/YPD. After additional shaking for 3 hr, cells were harvested as described for cultures in complex medium. Replicate cultures arise from independent main cultures.

## Perchloric-acid extraction of glucans and quantification

Unless otherwise noted, all steps were performed at 4°C. Yeast cell pellets from shake-flask experiments were resuspended in 3 volumes 1.12 M perchloric acid and 4 volumes glass beads (acid-washed, 425–600 µm diameter). To calculate the volume of the cell pellets, the wet weight [mg] was multiplied with the factor 0.9 [µL mg$^{-1}$]. The cells were then homogenized by vigorous vortexing for 45 to 50 min, and complete disruption of cells was confirmed at random using a light microscope. Samples were subject to centrifugation for 5 min at 6000 $g$, thereby fractionating the soluble fraction (supernatant; soluble glucans, sugars, water-soluble cell components) and the insoluble fraction (pellet; cell debris, insoluble glucans).

### Soluble fraction

The pH of the soluble fraction was adjusted to 6–7 with neutralization solution (2 M KOH, 0.4 M MES, 0.4 M KCl) and precipitated salts were removed by centrifugation for 5 min at 6,000 $g$. After taking an aliquot of the supernatant for glucan quantification, the remainder was stored at −20°C for glucan structural analyses (chain-length distributions). The aliquot used for glucan quantification was adjusted to 80% (v/v, final concentration) methanol and stored at −20°C for 16 hr to precipitate water-soluble polyglucans. These were pelleted by centrifugation for 5 min at 16,000 $g$. The resulting supernatant was transferred to a fresh tube for quantification of glucose and non-methanol precipitable glucans, i.e., malto-oligosaccharides, dried in a centrifugal evaporator and redissolved in water. The pellet (containing water-soluble, methanol-insoluble glucans, here named soluble glucans) was once washed with 75% (v/v) methanol, air-dried at 25°C and redissolved in water.

### Insoluble fraction

After removal of the soluble fraction, the glass beads were removed by repeatedly adding small volumes of water, vortexing the sample and transferring the suspension (containing insoluble glucans and cell debris but no glass beads) to a fresh tube until all the yeast material was transferred. After centrifugation at 6000 $g$ for 5 min, the supernatant was discarded and the pellet extensively washed with water until a neutral pH was reached. An aliquot from the freshly mixed suspension was taken for glucan quantification and the remainder stored at −20°C.

Soluble and insoluble glucans in the aliquots were quantified in an enzymatic assay after digestion to glucose as described in *Hostettler et al., (2011)*. The supernatant after methanol precipitation of the soluble fraction was measured once without prior digestion to glucose (giving the free glucose content) and once after digestion to glucose, and the malto-oligosaccharide content was calculated by subtracting free glucose from total glucan content in that fraction. The percentage of insoluble glucans from total glucans (i.e., sum of all insoluble and soluble glucans, including malto-oligosaccharides and free glucose) was calculated individually for each replicate. All yeast samples were included in the quantification, except for one culture of line 6 which failed induction (i.e. no glucans were made).

## Structural analyses of glucans

### Scanning electron microscopy (SEM)

Yeast cell pellets from liquid cultures with complex medium (inducing condition; grown as described above) were lysed for 3 hr with gentle shaking in 100 mM potassium phosphate buffer, pH 7.5, supplemented with *Arthobacter luteus* lyticase (500 U 100 mg$^{-1}$ cells [wet weight]). To ensure complete cell lysis, the samples were subject to centrifugation for 5 min at 6000 $g$, the supernatant discarded, the pellet frozen in liquid nitrogen, thawed and the digestion repeated. Complete lysis was confirmed by light microscopy. In addition, at least one negative control (yeast without insoluble glucans) was included in each preparation. Samples were then subjected to centrifugation at 6000 $g$ for 5 min, the pellets (containing insoluble glucans and cell debris) once washed with water and then resuspended in glucan buffer (50 mM Tris/HCl, pH 7.0, 0.2 mM EDTA, 0.5% [w/v] SDS). To separate insoluble glucans from cell debris, the suspension was sequentially laid onto two 100% Percoll cushions (GE Healthcare) and the insoluble glucans pelleted by centrifugation for 5 min at 2500 $g$ in a swing-out rotor. The glucan pellet was then treated with proteinase K for 16 hr at 37°C (1 µl 100 mg$^{-1}$ cells; Gateway LR Clonase II Enzyme Mix, Invitrogen) in glucan buffer supplemented with 3.2 mM CaCl$_2$ and purified on a third 100% Percoll cushion as before. The pellet containing the highly

pure insoluble glucans was washed three times with glucan buffer to remove proteinase K, then extensively in water to remove SDS, then cooled to 4°C until use.

For isolation of plant starch granules, rosettes from Arabidopsis wild-type plants (WS ecotype) were ground in liquid nitrogen. After resuspending the powder in starch buffer (50 mM Tris/HCl, pH 8.0, 0.2 mM EDTA, 0.5% [v/v] Triton X-100) and sequential filtering through 100 μm, 30 μm and 15 μm nylon nets, the suspension was laid onto a single 95% (5% [v/v] 0.5 M Tris-HCl, pH 8.0) Percoll cushion and starch granules pelleted by centrifugation for 20 min at 2500 *g*. The pellet was first once washed with 0.5% (v/v) SDS, then extensively in water, then cooled to 4°C until use.

For visualization, the glucans were coated with platinum (MED010, Balzers) and imaged with a Leo 1530 Gemini SEM (Hitachi, Düsseldorf, Germany) using a 7-kV electron beam.

## Cryo X-ray ptychographic tomography

Yeasts were grown in complex medium with galactose (inducing condition) as described above. After 6 hr incubation of the main culture, cells were harvested by centrifugation for 3 min at 500 *g* at 25°C, resuspended in 100 mM MES buffer, pH 6.5, then kept on ice until use (2 to 3 hr). Before injecting cells into the tips of glass micro-capillaries, glycerol was added to the cell suspension as cryo-protectant to a final concentration of ~17% (v/v). Purified, dry, amylose-free Arabidopsis starch granules (*gbss ptst* mutant starch) were resuspended in water 12 hr before injection into the capillary tips. Filled capillary tips were flash frozen in liquid ethane and stored briefly in liquid nitrogen before mounting for tomography measurements.

Cryo X-ray ptychographic tomography measurements (*Diaz et al., 2015*) were performed at the cSAXS beamline of the Swiss Light Source (Paul Scherrer Institut, Villigen, Switzerland) using a photon energy of 6.2 keV, corresponding to a wavelength of 0.2 nm. The beam was defined by a coherently illuminated Fresnel zone plate of 200 μm diameter and 90 nm outer-most zone width, providing a flux of about $4 \times 10^8$ photons s$^{-1}$. The specimens were mounted on a 3D piezoelectric positioning stage on top of a rotation stage 3.7 mm downstream of the focus, where the beam had a diameter of about 8.2 μm. Coherent diffraction patterns from the specimen were recorded with an Eiger 500 k detector with 75 μm pixel size (*Dinapoli et al., 2011*) placed 7.514 m downstream the sample position. For both specimens, ptychographic scans were acquired by scanning the sample in a grid following a Fermat spiral pattern (*Huang et al., 2014*) with an average step size of 2.5 μm over a field of view of $38 \times 22$ μm$^2$ (horizontal×vertical), corresponding to 131 points per scan. At each scanning position, diffraction patterns were recorded with an exposure time of 0.1 s. Ptychographic scans were repeated at 192 rotation angles of the specimen with respect to the beam ranging from 0 to 179.0625° with a step size of 0.9375°. During the entire acquisition, a cryo jet was blowing nitrogen gas at 110 K onto the glass capillary, ensuring that the frozen specimen remained in the vitreous phase for the measurements. Ptychographic reconstructions were performed using 200 iterations of a difference map algorithm (*Thibault et al., 2009*) followed by 100 iterations of a maximum likelihood refinement (*Thibault and Guizar-Sicairos, 2012*). Using $500 \times 500$ pixels of the detector to crop the diffraction patterns, the resulting images had a square pixel size of 40.07 nm. All reconstructed phase images from different angular positions were processed and combined in a tomographic reconstruction as described in *Guizar-Sicairos et al., (2011)*. For each sample a 3D resolution of about 150 nm was estimated by Fourier shell correlation of two tomograms, each of them obtained from half of the angular projections, using the half-bit threshold criterion (*Van Heel and Schatz, 2005*). The dose imparted on each specimen was estimated to be about $8 \times 10^6$ Gy.

For measurements of electron densities of insoluble glucans, 6 cubic regions with $20^3$ to $30^3$ voxels were randomly selected within a single tomogram among regions of high density (presumably constituting the insoluble glucans) from individual Arabidopsis starch granules or individual cells of line 29, respectively. For each selected region, the electron density value at the maximum of the histogram (i.e., the mode) was determined. The mean and standard deviation were calculated from values of the six cubic regions. Average electron densities were converted to mass densities (*Diaz et al., 2012*) using an A/Z ratio (mass number/atomic number) of 1.863, i.e., assuming a glucose polymer (formula $C_6H_{10}O_5$) with 27% (w/w) hydration, as suggested for B-type starch (*Pérez and Bertoft, 2010*). Note that the exact water content cannot be deduced from the measured density, but influences the conversion from measured electron density to final mass density only little, since the calculated A/Z ratio of water (A/Z = 1.80) and of glucan (A/Z = 1.89) are similar

(e.g., within the range of 10% to 35% [w/w] water content, the A/Z ratio and thus the calculated mass density differ by <1.2%). The mean electron densities ± S.D. calculated from the 6 cubic regions were $0.441 \pm 0.001$ e$^-$Å$^{-3}$ (corresponding to a mass density of $1.364 \pm 0.004$ g ml$^{-1}$) for Arabidopsis starch and $0.44 \pm 0.01$ e$^-$Å$^{-3}$ (corresponding to a mass density of $1.36 \pm 0.03$ g ml$^{-1}$) for insoluble glucans from yeast line 29. We additionally estimate a systematic error of ~0.003 e$^-$Å$^{-3}$ (corresponding to ~0.01 g ml$^{-1}$) in the absolute electron density measurement caused by noise in the tomographic reconstructions. The error given in the results section is the sum of this systematic error and the S.D.

## Small and wide-angle X-ray scattering (SAXS and WAXS)

Insoluble glucans were purified as described for SEM, but additionally washed in 80% (v/v) ethanol and desiccated under low pressure for two days. The powder was used directly for WAXS measurement. For SAXS measurements, a ~10% [w/w] glucan suspension in water was injected into capillaries, let stand until the glucans had settled, and diffraction patterns were acquired of glucan-enriched regions. SAXS and WAXS experiments were performed using a Rigaku MicroMax-002$^+$ microfocused beam (40 W, 45 kV, 0.88 mA) with the $\lambda_{CuK\alpha}$ = 0.15418 nm radiation in order to obtain direct information on the scattering patterns. The scattering intensities were collected by a Fujifilm BAS-MS 2025 imaging plate system (15.2 cm × 15.2 cm, 50 μm resolution) and a 2D Triton-200 X-ray detector (20 cm diameter, 200 μm resolution). An effective scattering vector range of 0.05 nm$^{-1}$ < q < 25 nm$^{-1}$ was obtained, where $q$ is the scattering wave vector defined as q = $4\pi \sin \theta / \lambda_{CuK\alpha}$ with a scattering angle of $2\theta$.

## Determination of chain-length distributions (CLDs) using high-performance anion-exchange chromatography with pulsed amperometric detection (HPAEC-PAD)

CLDs of soluble and insoluble glucans and malto-oligosaccharides were obtained from the perchloric-acid extracted glucans. Soluble glucans were first precipitated in methanol as described above, providing also the non-methanol precipitable glucans (malto-oligosaccharides or MOS). Arabidopsis leaf starch from wild-type (WS ecotype) and mutants was extracted using perchloric acid as described (*Pfister et al., 2014*). Soluble and insoluble glucan fractions were solubilized by heating to 99°C for 15 min. Samples were then debranched for 2.5 hr at 37°C in a reaction containing 1 U *Klebsiella planticola* pullulanase (M1, Megazyme), 0.04 U of *Pseudomonas* isoamylase (Megazyme) and 10 mM sodium acetate, pH 4.8. The reaction was stopped by heating to 99°C for 5 min. The samples were clarified by centrifugation for 1 min at 16,000 *g* and the supernatant applied to sequential 1.5 mL columns of Dowex 50WX4 (hydrogen form, 100–200 mesh) and Dowex 1 × 8 (chloride form, 200–400 mesh). The glucans were eluted in 4 mL water, lyophilized and re-dissolved in water by heating to 99°C for 5 min. In addition, non-debranched controls of soluble and insoluble glucans from several yeast lines were prepared by omitting the debranching enzymes. Malto-oligosaccharides were treated in the same way as these non-debranched controls. Chains were separated on a HPAEC-PAD system (ICS-5000, Dionex) using a CarboPack PA-200 column, a flow rate of 0.5 mL min$^{-1}$ and the following program: 0 to 12.5 min, a linear gradient from 95% eluent A (100 mM NaOH) and 5% B (150 mM NaOH, 0.5 M sodium acetate) to 60% A and 40% B; 12.5 to 50 min, a linear gradient to 15% A and 85% B; 50 to 52 min, a linear gradient to 10% A and 90% B; 52 to 60 min, 10% A and 90% B (column wash); 60 to 80 min, 95% A and 5% B (column equilibration) (all eluent percentages are given as volume per volume). Peaks were identified using standard linear glucan chains with degrees of polymerization (DP) from 2 to 7 (maltose to maltoheptaose). Peak areas were determined with Chromeleon software and summed within the depicted range (DP 3 to 60 in *Figure 5A*, and DP from 3 to 45 for the comparisons in *Figure 5B–D*). Non-debranched controls of soluble and insoluble glucans gave negligible peaks. CLDs from plant glucans in *Figure 5B and C* were recalculated from *Pfister et al., (2014)*. In the differences plots, the final error was calculated as the square root of the sum of the squared S.E.M.s. Very rarely, a replicate sample was excluded if it showed signs of degradation (specified by a low overall peak area relative to other sample replicates or obvious degradation products).

## Iodine absorption spectra

Iodine absorption spectra from soluble and insoluble glucans from yeast (from perchloric-acid extraction), Arabidopsis and potato amylopectin and potato amylose (see Chemicals, Media and Plant Materials) were obtained using an adaptation of a described method (*Krisman, 1962*). Glucan solutions (1.4 g L$^{-1}$ and 2.8 g L$^{-1}$ in water for insoluble and soluble glucans, respectively) were heated to 99°C for 10 min and then clarified by centrifugation at 16,000 *g* for 1 min. One volume of supernatant was mixed with 7 volumes of iodine/potassium iodide solution (2 M CaCl$_2$, 3.1 mM KI, 0.2 mM I$_2$), and the absorption in steps of 1 or 2 nm immediately measured. The spectrum from a water blank with iodine solution was subtracted from the glucan absorption spectrum. The absorptions in the wavelength range provided in the Figures (420 to 780 nm in *Figure 6C*; 450 to 650 nm in *Figure 6—figure supplement 2*) were normalised.

## Iodine staining and microscopy of yeast

### Iodine staining of yeast grown on plates

Cells from YPD plates were resuspended in water and dropped onto an SC-gal plate, which was then incubated at 30°C for 24 hr. Subsequently, native suspensions of amylose-free Arabidopsis and potato starch and pre-boiled potato amylose (type III) in water were dropped onto the plate. Once dry, the plate was exposed to vapor of Lugol's solution and immediately photographed.

### Light micrographs (LM)

Cells from liquid cultures in complex medium were mixed with one volume Lugol's solution and immediately imaged using an Axio Imager.Z2 light microscope equipped with a 100X oil-immersion objective (Zeiss, Oberkochen, Germany).

### Chemical fixation and transmission electron micrographs (TEM)

Cells grown in liquid cultures in complex medium (inducing condition, as described above) were pre-fixed with glutaraldehyde (50 mM sodium cacodylate, pH 6.8, 1 mM MgCl$_2$, 1 mM CaCl$_2$, 2% [w/v] glutaraldehyde; final concentrations) for 5 min, pelleted by centrifugation for 3 min with 1000 *g* and resuspended in glutaraldehyde fixative (final concentrations as above). After incubation for 16 hr at 4°C, cells were pelleted and washed four times with water, then once with 100 mM sodium cacodylate buffer, pH 6.8. Cells were post-fixed at 4°C with increasing concentrations of osmium tetroxide in 100 mM sodium cacodylate, pH 6.8, starting from 0.2% (w/v) osmium tetroxide for 5 hr, followed by 0.5% (w/v) for 16 hr and finally 1% (w/v) for 2 hr. Cells were pelleted by gentle centrifugation and washed three times with 100 mM sodium cacodylate, pH 6.8, at 4°C, then once with water at 25°C. Cell dehydration was performed by sequential washes in increasing ethanol concentrations (25%, 50%, 75%, 95%, 99.9%, all given as v/v), six washes in 100% ethanol and two washes in 100% acetone whereat each step had a total duration of 5–8 min including centrifugation. Finally, cells were infiltrated with epon (47.5% epon 812; 34.4% methylnadic anhydride [MNA], 16.4% dodecenyl succinic anhydride [DDSA], 1.7% DMP-30), by gradually increasing its concentration in acetone, beginning with 25% (v/v) for 2 hr, then 50% (v/v) for another 2 hr, followed by 75% (v/v) for 16 hr and five times 100% for a total period of 7 hr. Finally, cells were transferred to gelatin capsules filled with freshly prepared 100% epon and incubated at 60°C for 48 hr for the polymerization of the embedding medium.

Ultrathin (70 nm) sections were cut with a diamond knife, placed on formvar/carbon-coated copper grids, stained with 2% (w/v) uranyl acetate and Reynold's lead citrate. Images were acquired using a FEI Morgagni 268 TEM.

## Native page and western blotting

For isolation of soluble proteins, yeast cell pellets from liquid cultures with complex medium were resuspended in 3.3 volumes of native extraction buffer (100 mM MOPS, pH 7.5, 1 mM EDTA, 5 mM DTT, 10% [v/v] glycerol, protease inhibitor [Complete EDTA-free; Roche]) and 4.3 volumes of glass beads (acid-washed, 425–600 μm diameter) and homogenized by vortexing for 6 min in total at 4°C with cooling in between. Rosette material from two pooled four-week-old Arabidopsis plants (harvested in the middle of a 12 hr day) was homogenized in the same extraction buffer as above in an all-glass homogenizer. Soluble proteins were separated from cell debris by two sequential

centrifugation steps (each 8 min, 16,000 g at 4°C), frozen in liquid nitrogen and stored at −80°C until use. Protein amounts were determined using a Bradford-based protein assay (Bio-Rad) with bovine serum albumin (BSA) as standard. Equal loading was confirmed in SDS-PAGE.

For total protein extracts, yeasts were grown and homogenized as above, but with non-native extraction buffer (100 mM Tris, pH 7.0, 2% [w/v] SDS, protease inhibitor [Complete EDTA-free; Roche]). The extracts were heated to 99°C for 12 min, then clarified by centrifugation for 5 min at 16,000 g at 25°C. The supernatant, constituting the total protein extract, was frozen in liquid nitrogen and stored at −80°C until use. Protein concentrations were measured using the bicinchoninic acid (BCA) protein assay (Pierce) with BSA as standard.

To monitor branching enzyme and isoamylase activity, 15 μg of native soluble proteins were loaded on a 7.5% (w/v) native polyacrylamide gel containing 0.015% (w/v) oyster glycogen and run at 10 V cm$^{-1}$ for 3.5 hr at 4°C. After washing for 30 min at 4°C in 50 mM Hepes-NaOH, pH 7.0, and 10% (v/v) glycerol, the gel was incubated overnight at 25°C with gentle shaking in 50 mM Hepes-NaOH, 10% (v/v) glycerol, 2.5 mM AMP, 50 mM glucose 1-phosphate and 28 U (per gel) phosphorylase a (from rabbit muscle).

For starch synthase activity, 22.5 μg of proteins were separated on a 7.5% (w/v) native polyacrylamide gel containing 0.3% (w/v) oyster glycogen using 10 V cm$^{-1}$ for 3.5 hr at 4°C. The gels were incubated for 16 hr at 25°C with gentle shaking in 100 mM HEPES-NaOH, pH 7.5, 2 mM DTT, 10% (v/v) glycerol, 0.5 mM EDTA, 0.5 M trisodium citrate (tribasic) and 0.8 mM ADPglucose. Both types of gels were stained with ~1:3 diluted Lugol's solution to visualize glucan-modifying enzyme activities.

For the immunoblots shown in *Figures 2B* 15 μg of either soluble or total proteins were loaded on 12.5% (w/v) (for SS2 and GlgC-TM) or 7.5% (w/v) (for SS4) SDS-PAGE gels, blotted on polyvinylidene difluoride membranes (Immobilon-P, Millipore) and probed with monoclonal anti-HA-peroxidase (clone HA-7, H6533; RRID:AB_439706) (*Egan et al., 2014*) or anti-FLAG (M2, F1804; RRID: AB_262044) primary antibody (an antibody validation profile can be found in 1DegreeBio under http://1degreebio.org/reagents/product/751433/?qid=1013436; accessed 1.11.2015) and goat anti-mouse IgG (H + L)-HRP secondary antibody (BIO-RAD, #1706516), respectively. Chemiluminescence was visualized using the WesternBright ECL HRP substrate kit (Advansta). The immunoblot shown in *Figure 2—figure supplement 1C* was performed using total protein extracts as the described above for SS4, but using 7.5 μg of total proteins.

## Statistical methods

Data analyzed by t tests or ANOVA (*Figure 3—figure supplement 2* and *Figure 4A,B*) passed the Shapiro-Wilk test for normality (p value $\geq$ 0.05). Equality of variances between compared samples was assessed using F-tests (α = 0.05). Pair-wise comparisons of glucan content (*Figure 3—figure supplement 2*) and of untransformed percentages of insoluble glucans (*Figure 4A*) were performed using two-sided t-tests (with Welch's correction when equal variances could not be assumed). Multiple comparisons of untransformed percentages of insoluble glucans (*Figure 4B*) were performed using one-way ANOVA with Tukey multiple comparison test as equal variances among compared samples could be assumed.

## Accession numbers

The Arabidopsis Genome Initiative gene codes for the Arabidopsis genes used in this study are the following: *ISA1*, At2g39930; *ISA2*, At1g03310; *SS1*, At5g24300; *SS2*, At3g01180; *SS3*, At1g11720; *SS4*, At4g18240; *BE1*, At3g20440; *BE2*, At5g03650; *BE3*, At2g36390. The GenBank accession number for *glgC*, the AGPase from *Escherichia coli*, is V00281.1. The gene IDs of the *S. cerevisiae* loci other than the loci of the yeast expression platform (*Mikkelsen et al., 2012*) are as follows: *GSY1*, YFR015C; *GSY2*, YLR258W; *GLC3*, YEL011W; *GLG1*, YKR058W; *GLG2*, YJL137C; *GDB1*, YPR184W; *GPH1*, YPR160W (CENPK113-7D database; www.sysbio.se/cenpk).

## Acknowledgements

We thank Barbara A. Halkier and Bo Salomonsen for kind donation of yeast CEN.PK113-11C, yeast integration vectors and detailed protocols. We are grateful to Gebhard Schertler, Takashi Ishikawa, and Andreas Menzel for the collaboration with Paul Scherrer Institut, Tom Okita and Sam Mugford

for providing a plasmid with *glgC-TM*, David Seung and Richard Visser for provision of Arabidopsis and potato starch, respectively, Biljana Didic for preparation of media, Simona Eicke and the microscopy facility of ETH Zurich (ScopeM) for help with microscopy, Martha Stadler-Waibel for technical help, Michaela Stettler and Sebastian Streb for sharing expertise on HPAEC-PAD, Mario Coiro for advice on statistics, and Alison Smith for valuable suggestions on the manuscript.

## Additional information

### Funding

| Funder | Grant reference number | Author |
|---|---|---|
| Schweizerischer Nationalfonds zur Förderung der Wissenschaftlichen Forschung | 31003A_153144 | Samuel C Zeeman |
| Schweizerischer Nationalfonds zur Förderung der Wissenschaftlichen Forschung | 31CP30_163503 | Samuel C Zeeman |

The funders had no role in study design, data collection and interpretation, or the decision to submit the work for publication.

### Author contributions

BP, Conception and design, Acquisition of data, Analysis and interpretation of data, Drafting or revising the article; AS-F, AD, KL, FM, Acquisition of data, Analysis and interpretation of data, Drafting or revising the article; CO, MH, FRS, Acquisition of data, Analysis and interpretation of data; RM, Analysis and interpretation of data, Drafting or revising the article; SCZ, Conception and design, Analysis and interpretation of data, Drafting or revising the article

### Author ORCIDs

Samuel C Zeeman, http://orcid.org/0000-0002-2791-0915

## Additional files

### Supplementary files

• Supplementary file 1. Tables of the yeast strains (A), plasmids (B), primers for cloning (C), primers for genotyping (D) and of the plant lines (E) used in the present study.

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
