## [Decision Letter]

Thank you for submitting "Recreating the Synthesis of Starch Granules in Yeast" for consideration by *eLife*. Your paper has been reviewed by three peer reviewers, and the evaluation has been overseen by a Reviewing Editor and Detlef Weigel as the Senior Editor.

The reviewers have discussed their comments with one another, and, based on the individual reviews and the reviewers' discussion, the Reviewing Editor has drafted this decision to help you prepare a revised submission.

Summary:

Starch is the most important storage carbohydrate in plants, with key roles in plants and serving as the major dietary source of carbohydrates for humans. It l also has many non-food applications. Differences in starch structure between and within species are of great biological, nutritional and technical importance. Starch is also immensely complex, consisting of an unbranched glucan (amylose) and a branched glucan (amylopectin), which form a semi-crystalline starch granule. This physicochemical complexity sets starch apart from all other storage glucans, including the widely distributed soluble glycogen. Starch structure and the role of different enzymes in generating this structure have been intensively studied in several plant species including the referenced system Arabidopsis and also many crops including maize and potato. This has occurred out of curiosity, but also frequently driven by possible applications. The approach has mainly involved using forward or reverse genetics, followed by chemical and physical analysis of the resulting changes in starch structure. These studies have defined a set of enzymes that are involved in starch synthesis – or more precisely, whose modification / deletion alters the levels or especially the structure of starch. These genes are being used to try to modify starch structure in Arabidopsis and several crop plants.

However, although very successful, this previous research does not provide a full understanding of how starch is synthesized, and the structure of starch and starch granules. This requires the reconstitution of the entire system from defined parts, i.e. synthetic biology.

This is the potentially game-changing advance that is reported by Zeeman and colleagues.

Starting with a yeast strain in which all glycogen-metabolizing genes are deleted, the individual components can be added all together, or in combinations or alone to test how a starch-like structure can be created. This paper convincingly shows that each of the individual enzyme is active, and that when all are present they form a starch-like structure which is analyzed chemically and physically to show the properties are much more similar to starch from Arabidopsis than to glycogen.

Specifically, using a combinatorial approach, the authors examined nearly 30 recombinant yeast strains containing plant starch synthases (SS), branching enzymes (BE), and a debranching isoamylase heterodimer (ISA). A particular strain ("line 29") expressing four SS, two BE, and the ISA heterodimer, along with an ADP-glucose pyrophosphorylase, was able to recapitulate the abundant production of insoluble starch granules with a structure highly similar to those from wild-type *A. thaliana*. Numerous other strains produced α-glucans of distinct primary structure and morphology.

The authors also convincingly argue why this is best done is yeast, being faster and more generalizable. Studies of starch structure in mutants are unavoidably complicated by the simultaneous presence of starch degrading enzymes that may – and almost certainly do – act on incorrectly formed starch. The use of a completely heterologous system allows to rigorously dissect how starch and starch grains are formed.

As such, this work demonstrates and validates a powerful new model system to dissect the complex, crucially topical process of starch biosynthesis across plant species. This system also has the potential to explore the effects of protein mutants/variants, post-translational modifications, and cooperativity with additional proteins and enzymes. The study appears to be technically sound and performed with a high degree of rigor.

In short, this research is a tour-de-force and opens new perspectives for understanding how starch is synthesized and using this knowledge to modify starch structure.

Essential revisions:

1) As mentioned at the end of the Discussion, starch grains contain amylopectin and amylose and the latter is formed by GBSS. In this study, GBSS was omitted, and the focus was on formation of an amylopectin-like macromolecule. This does need to be justified. It may be important to compare the amylopectin-like structure formed in yeast with the structure of amylopectin in gbss1 mutants (is it identical with amylopectin in wild-type Arabidopsis). This can probably be done very easily, probably even using information available in the literature.

2) Consideration should be given to why the repeat structure of the insoluble glucan in strain 29 is not the same as for Arabidopsis starch. The fact that the enzymic reconstruction omitted one branching enzyme (BE1) should be discussed. Although the authors have published that knock out of this gene has no discernible phenotype in Arabidopsis, it remains possible that in the yeast system this could make an important contribution to starch structure.

3) It would be helpful to have experimental data for results in yeast that were not predicted by the work from plants. For example, there are examples within the yeast system where insoluble glucan is produced in significant amounts (Figure 3—figure supplement 3 line I) which contradict some of the basic principles that have emerged from the plant work (SS2 plus BE3 only without ISA) which are not discussed properly nor is any analysis of the resulting glucan structure offered to illuminate this finding. This result contradicts the authors' claim in the Discussion that all SSs can generate soluble glucans in combination with BEs and ISA – clearly at least one SS can do this without ISA. Outside of this exception, the importance of ISA activity in insoluble glucan formation is not discussed fully except in the context of removing soluble glucan.

4) The analysis of Arabidopsis mutants has suggested that the role of SS4 in starch biosynthesis is due to its ability to prime granule formation. However, the yeast results suggest that SS3 and possibly SS2 might also be able to prime insoluble glucan accumulation. Some discussion of these results needs be offered as well as comparison of the 'starch structure' between lines 5, 7 and 29 to compare the arrangements of glucan formed by limited complements of enzymes compared to the full complement in line 29.

5) Looking at Figure 4, it might be pointed out that the DP>18 are strongly overrepresented in the yeast product compared to Arabidopsis. Is there an explanation for this?

6) In the first sentence of the Discussion, 'genuine' starch granules may be an overstatement, especially as GBSS and presumably amylose are missing. Further on in the same paragraph, 'the highest degree of similarity' can perhaps be rephrased – what the authors mean is probably that the glucan was most similar to those from Arabidopsis when all enzymes were included, but this sentence could also be read otherwise.

7) Some of the discussion /comparison of the results in yeast and plants that is at the moment in the Results should be moved to the Discussion. Splitting it between the Results and Discussion does not help to bring over the main messages. It also leaves the Discussion a little short on new insights and a little long on what might be done in the future. Also, more discussion of the roles on glycogenines in yeast, and the ability of the cocktail of enzymes to create a starch-like structure without a seeding polysaccharide or glycoprotein might be added.

8) The authors have put much emphasis on the results from strain 29, which carries genes encoding the entire complement of starch biosynthetic enzymes defined in Arabidopsis (except GBSS and SBE1). However, the presentation of their analysis might be improved by comparing and contrasting insoluble glucan made in Arabidopsis and yeast, with additional data on the structure of the insoluble glucans produced in yeast in strains where such production was unexpected. An analysis of what did not work as expected rather that only what did work, or a systematic discussion of the contribution of each component in isolation and in combination could provide a better interpretation of the data already generated as well as establishing how robust the yeast model is for starch synthesis in plants.

9) For a general reader, a brief account of the colors obtained with iodine with different glucans (including glycogen) would be very helpful.

10) Similarly the method used to separate soluble and insoluble glycans should be briefly explained in the text.

11) The authors should include a precise description of the glycosidic linkages of amylose and amylopectin in the second paragraph of the Introduction.

---

## [Author Response]

[…]

*Essential revisions:*

*1) As mentioned at the end of the Discussion, starch grains contain amylopectin and amylose and the latter is formed by GBSS. In this study, GBSS was omitted, and the focus was on formation of an amylopectin-like macromolecule. This does need to be justified. It may be important to compare the amylopectin-like structure formed in yeast with the structure of amylopectin in gbss1 mutants (is it identical with amylopectin in wild-type Arabidopsis). This can probably be done very easily, probably even using information available in the literature.*

The amylopectin chain-length distribution is not altered upon loss of GBSS in Arabidopsis (e.g. Seung et al. 2016; Supplemental Figure S7). Likewise, the amylopectin from wild-type Arabidopsis starch (separated from amylose by size-exclusion chromatography) has a wavelength of maximal absorption (λ_max_) of 550 nm after complexion with iodine (Zeeman et al., 2002), very close to that of *gbss ptst* amylopectin (548 nm). These points are now mentioned in the manuscript. We used starch from the Arabidopsis *gbss ptst* mutant (lacking the only GBSS isoform and its starch-targeting protein, Seung et al. 2015) in addition to wild-type starch for structural analyses (SEM, SAXS, WAXS) where comparisons had not previously been made. We are confident that the absence of GBSS (and PTST) does not alter amylopectin structure in Arabidopsis mutants.

*2) Consideration should be given to why the repeat structure of the insoluble glucan in strain 29 is not the same as for Arabidopsis starch. The fact that the enzymic reconstruction omitted one branching enzyme (BE1) should be discussed. Although the authors have published that knock out of this gene has no discernible phenotype in Arabidopsis, it remains possible that in the yeast system this could make an important contribution to starch structure.*

Most plant species contain two classes of branching enzymes (class I and II). In Arabidopsis, however, both BE2 and BE3 fall into class II, and there is no class I branching enzyme (Dumez et al., 2006). The gene annotated as BE1 belongs to the putative class III of branching enzymes (Han et al. 2007). However, no branching enzyme activity or role in starch metabolism has been reported for this or any other enzyme of the putative class. The *be1* mutant of Arabidopsis was first reported to have a wild-type phenotype (Dumez et al. 2006), while the *be2be3* double mutant was reported to have no starch synthesis (i.e. a BE null mutant). Later, the *be1* mutant was reported to be embryo lethal (Wang et al. 2010). We nevertheless agreed with the reviewers that it was interesting to test the function of BE1 in the yeast system. We therefore expressed it in two different backgrounds; 1) together with SS3 and the AGPase (strains V and W, new Figure 2—figure supplement 2). This strain only contained amylose-like glucans like its parental line with just SS3 and the AGPase, indicating that BE1 alone has little or no branching activity. Second, we included BE1 in strain 29 (i.e. in addition to the other amylopectin-synthesizing enzymes). Here again, BE1 did not alter glucan production or structure (as judged by iodine staining, chain-length distribution, granule morphology, X-ray scattering patterns; new Figure 2—figure supplement 2 and data not shown). Thus, the absence of BE1 cannot explain the atypical repeat structure in the insoluble glucans from strain 29. Rather we believe that the repeat size stems from an enzymatic imbalance; more specifically excessive synthase activity relative to branching activity – a hypothesis we state clearly in the Discussion of the revised manuscript (section “Current limitations and future applications of the yeast system”) and which we will test in future research.

*3) It would be helpful to have experimental data for results in yeast that were not predicted by the work from plants. For example, there are examples within the yeast system where insoluble glucan is produced in significant amounts (Figure 3—figure supplement 3 line I) which contradict some of the basic principles that have emerged from the plant work (SS2 plus BE3 only without ISA) which are not discussed properly nor is any analysis of the resulting glucan structure offered to illuminate this finding. This result contradicts the authors' claim in the Discussion that all SSs can generate soluble glucans in combination with BEs and ISA – clearly at least one SS can do this without ISA. Outside of this exception, the importance of ISA activity in insoluble glucan formation is not discussed fully except in the context of removing soluble glucan.*

The reviewers raise an important point here: the production of insoluble glucans by SS2 in the absence of ISA is interesting. In the revised manuscript, we have investigated this aspect systematically for all SSs. For this we created an additional set of yeast strains expressing only BE3 (like in line I) instead of both BE2 and BE3 (revised Figure 4). These data show that SS3 is also capable of producing insoluble glucans in the absence of ISA, and that in both instances this is dependent on having only BE3 instead of both BE2 and BE3. Iodine staining of the glucans revealed a consistent shift towards higher λ_max_ values (wavelength of maximum absorption) when only BE3 is present. This indicates that more chains are able to form secondary structures, which presumably renders part of the glucans insoluble, despite the absence of ISA. We now discuss this finding and the role of isoamylase during the formation of insoluble glucans more extensively in the Discussion section “Role of ISA, SS isoforms and BEs during the formation of insoluble glucans”.

*4) The analysis of Arabidopsis mutants has suggested that the role of SS4 in starch biosynthesis is due to its ability to prime granule formation. However, the yeast results suggest that SS3 and possibly SS2 might also be able to prime insoluble glucan accumulation. Some discussion of these results needs be offered as well as comparison of the 'starch structure' between lines 5, 7 and 29 to compare the arrangements of glucan formed by limited complements of enzymes compared to the full complement in line 29.*

In the revised manuscript we now distinguish more clearly the priming of granule formation (i.e. whether a given SS can efficiently use available primers or generate its own) and whether the glucan product made by the concerted action of a given SS, BE and ISA combination can crystallize.

We therefore investigated the capability of each SS isoform to initiate glucans in the absence and presence of the yeast’s endogenous glycogenins (new Figure 3—figure supplement 2; discussed in section “Seeding of glucan synthesis by SS isoforms”). This shows that SS1 and – to a lesser extent – SS2 are partly dependent on glycogenins for glucan priming, meaning that they are less efficient in using available primers or generating suitable primers on their own. We acknowledge that we don’t know what is used as a primer in the absence of the glycogenins.

We also added a more detailed discussion on the role of each enzyme for the formation of an insoluble glucans (Discussion section “Role of ISA, SS isoforms and BEs during the formation of insoluble glucans”), as described in review point 3.

*5) Looking at Figure 4, it might be pointed out that the DP>18 are strongly overrepresented in the yeast product compared to Arabidopsis. Is there an explanation for this?*

We believe that the overrepresentation of DP > 18 goes along with the altered width of the lamellar repeat and is due to an imbalance in enzyme activity (too much chain elongation by SS3 and/or SS4 for the branching activity present; as described in our answer to point 3). The overrepresentation of DP>18 is mentioned in the Results section “Yeast glucans have semi-crystalline properties of starch” and in the Discussion section “Current limitations and future applications of the yeast system”. There is also references given in that section to a paper where plant BE mutants were shown to have a longer lamellar repeat than normal. We argue that this can be explained by our imbalance theory.

*6) In the first sentence of the Discussion, 'genuine' starch granules may be an overstatement, especially as GBSS and presumably amylose are missing. Further on in the same paragraph, 'the highest degree of similarity' can perhaps be rephrased – what the authors mean is probably that the glucan was most similar to those from Arabidopsis when all enzymes were included, but this sentence could also be read otherwise.*

We have changed “genuine” starch granules to “insoluble, starch-like granules” and rephrased the paragraph accordingly.

*7) Some of the discussion /comparison of the results in yeast and plants that is at the moment in the Results should be moved to the Discussion. Splitting it between the Results and Discussion does not help to bring over the main messages. It also leaves the Discussion a little short on new insights and a little long on what might be done in the future. Also, more discussion of the roles on glycogenines in yeast, and the ability of the cocktail of enzymes to create a starch-like structure without a seeding polysaccharide or glycoprotein might be added.*

We have searched the manuscript for inappropriate discussion in the results and moved these to the Discussion.

We have also included the analysis of additional yeast strains still expressing the glycogenins in the manuscript (new Figure 3—figure supplement 2), which shows that SS1 and – to a lesser extent – SS2 are partly dependent on glycogenins for glucan priming, as mentioned above. We discuss this aspect and the seeding of glucan synthesis now in the Discussion section “Seeding of glucan synthesis by SS isoforms”.

*8) The authors have put much emphasis on the results from strain 29, which carries genes encoding the entire complement of starch biosynthetic enzymes defined in Arabidopsis (except GBSS and SBE1). However, the presentation of their analysis might be improved by comparing and contrasting insoluble glucan made in Arabidopsis and yeast, with additional data on the structure of the insoluble glucans produced in yeast in strains where such production was unexpected. An analysis of what did not work as expected rather that only what did work, or a systematic discussion of the contribution of each component in isolation and in combination could provide a better interpretation of the data already generated as well as establishing how robust the yeast model is for starch synthesis in plants.*

As described above (point 3), we have added a systematic analysis of the “unexpected” production of insoluble glucans, i.e. in the absence of isoamylase (revised Figure 4), together with iodine staining characteristics. These data show that, in addition to isoamylase and starch synthases, also the level of branching activity influences whether or not insoluble glucans are produced. We now describe the contribution of each of these factors in the Discussion (section “Role of ISA, SS isoforms and BEs on the formation of insoluble glucans”), describing both what is consistent with previous plant work and what is not.

*9) For a general reader, a brief account of the colors obtained with iodine with different glucans (including glycogen) would be very helpful.*

We have added an explanation to the Results section (“Yeast strains produce high amounts of glucans”).

*10) Similarly the method used to separate soluble and insoluble glycans should be briefly explained in the text.*

We have added an explanation to the Results section (“Yeast strains produce high amounts of glucans”).

*11) The authors should include a precise description of the glycosidic linkages of amylose and amylopectin in the second paragraph of the Introduction.*

We have added a precise description in the second paragraph of the Introduction.